# Piezo's membrane footprint and its contribution to mechanosensitivity

Christoph A Haselwandter[1,2†*], Roderick MacKinnon[3†*]

[1]Department of Physics & Astronomy, University of Southern California, Los Angeles, United States; [2]Department of Biological Sciences, University of Southern California, Los Angeles, United States; [3]Laboratory of Molecular Neurobiology and Biophysics, Howard Hughes Medical Institute, The Rockefeller University, New York, United States

**Abstract** Piezo1 is an ion channel that gates open when mechanical force is applied to a cell membrane, thus allowing cells to detect and respond to mechanical stimulation. Molecular structures of Piezo1 reveal a large ion channel with an unusually curved shape. This study analyzes how such a curved ion channel interacts energetically with the cell membrane. Through membrane mechanical calculations, we show that Piezo1 deforms the membrane shape outside the perimeter of the channel into a curved 'membrane footprint'. This membrane footprint amplifies the sensitivity of Piezo1 to changes in membrane tension, rendering it exquisitely responsive. We assert that the shape of the Piezo channel is an elegant example of molecular form evolved to optimize a specific function, in this case tension sensitivity. Furthermore, the predicted influence of the membrane footprint on Piezo gating is consistent with the demonstrated importance of membrane-cytoskeletal attachments to Piezo gating.
DOI: https://doi.org/10.7554/eLife.41968.001

*For correspondence:
cah77@usc.edu (CAH);
mackinn@mail.rockefeller.edu
(RM)

†These authors contributed
equally to this work

**Competing interests:** The
authors declare that no
competing interests exist.

**Reviewing editor:** Baron
Chanda, University of Wisconsin-
Madison, United States

## Introduction

Piezo ion channels transduce mechanical stimuli into electrical activity (*Coste et al., 2010*). These channels – Piezo1 and Piezo2 in mammals – underlie many important processes in biology, including cell volume regulation in erythrocytes, cardiovascular system development, and touch sensation (*Maksimovic et al., 2014*; *Ranade et al., 2014a*; *Ranade et al., 2014b*; *Cahalan et al., 2015*). In electrophysiological experiments Piezo channels seem to be exquisitely sensitive to applied mechanical force: when the membrane of a cell is poked gently with a probe, or when pressure is applied to stretch a small patch of cell membrane on a gigaseal pipette, Piezo channels open (*Coste et al., 2010*; *Lewis and Grandl, 2015*).

Studies have addressed how Piezo channels 'sense' and open in response to mechanical force. In one approach Piezo channels, purified and reconstituted into droplet bilayers, opened when force was applied by swelling a droplet (*Syeda et al., 2016*). This observation implies that Piezo needs only the cell membrane to couple mechanical forces to pore opening. In another approach, Piezo channels in patches excised from cell membrane blebs (*Cox et al., 2016*), or in cell excised patches with applied positive or negative pressure (*Lewis and Grandl, 2015*), open in response to pressure application. These observations also support the notion that Piezo only needs an intact lipid membrane to transduce force into pore opening.

The reconstitution (*Syeda et al., 2016*) and excised patch (*Lewis and Grandl, 2015*; *Cox et al., 2016*) experiments suggest the 'force-from-membrane' hypothesis for mechanosensitive gating, which, in its simplest form, invokes lateral membrane tension as the origin of the 'opening force' (*Sukharev et al., 1999*; *Perozo et al., 2002*; *Chiang et al., 2004*; *Teng et al., 2015*). But other experiments suggest additional possibilities for force exertion. When blebs are formed on the

surface of a cell by removing local cytoskeletal attachments, certain properties of Piezo mechanosensitive gating change (*Cox et al., 2016*). And more directly, Piezo gating is altered by applying force to a tether artificially attached to the channel (*Wu et al., 2016*). Therefore, while membrane-mediated forces alone appear to be sufficient to open Piezo, tethers attached to the membrane or to the channel itself also seem to play a role in Piezo gating.

A partial molecular structure of a Piezo channel has been determined (*Guo and MacKinnon, 2017*; *Saotome et al., 2018*; *Zhao et al., 2018*). Piezo is a trimer of 3 identical subunits that form one central pore and three long arms that extend away from the center. A peculiar aspect of the structure is that the extended arms, which are made of transmembrane helices, do not lie in a plane as would be expected if Piezo normally resides in a planar membrane like most other ion channels. This property of the structure implied that Piezo likely curves the cell membrane locally into a spherical dome (projecting into the cell), which was confirmed by electron micrographs of small unilamellar lipid vesicles (*Guo and MacKinnon, 2017*).

On the basis of Piezo's demonstrated ability to curve lipid membranes locally into a dome, a mechanism for membrane tension sensitivity – called the membrane dome mechanism – was proposed (*Guo and MacKinnon, 2017*). Simply stated, the dome shape provides a source of potential energy for gating – in the form of excess membrane area 'stored' by curving the membrane – when the membrane comes under tension. If the Piezo dome becomes flatter when Piezo opens, then the projected (in-plane) area of the dome will expand, that is, the available in-plane area of the membrane-Piezo system will increase. Under tension $\gamma$, the flatter shape will be favored by energy $\gamma \Delta A$, where $\Delta A$ is here the change in the projected area of the Piezo dome. Therefore, this model rationalized Piezo's peculiar shape as a means to utilize, for gating purposes, the energy stored in a curved membrane under tension.

However, the membrane dome model of Piezo gating only considered the shape of the membrane within Piezo's perimeter and not the shape of the surrounding membrane, which is necessarily coupled to the curvature of the Piezo dome. In the present analysis we study the energetic contribution to Piezo gating provided by the shape of the surrounding membrane. Through membrane mechanical calculations, we show that the Piezo dome can strongly curve the surrounding membrane. We find that the energetic coupling between the shape of the Piezo dome and the surrounding membrane amplifies Piezo's tension sensitivity, and may explain the experimentally observed regulation of Piezo gating by membrane-cytoskeletal attachments.

## Results

### System of Piezo plus membrane

*Figure 1A and B* show two orientations of the molecular model of Piezo1 in yellow, which from here on we refer to as Piezo. Shown in grey, a spherical cap is placed such that it intersects the protein near the middle of the transmembrane helices. This grey surface therefore corresponds to the mid-bilayer surface of the membrane. We call the grey spherical cap, with its embedded Piezo channel, a mid-bilayer representation of the Piezo dome. This dome shape, produced by curved Piezo channels embedded in lipid bilayer membranes, has been confirmed experimentally (*Guo and MacKinnon, 2017*). The intersection of the grey surface and the Piezo channel, shown in cyan, informs that the dome surface area is covered by approximately 20% protein and 80% lipid membrane. Note that, if the unperturbed configuration of the lipid membrane is planar, the Piezo protein must apply, through its curved structure, a distorting force on the membrane to locally bend the membrane into a dome shape. And, of course, the membrane applies an opposing force on the protein. The result is a stable, non-planar equilibrium configuration of the membrane-Piezo system with zero net force, in which the sum energy of the channel and the membrane is minimized. In the present analysis we do not consider the flexing of Piezo. Instead, we focus on the membrane shape associated with a particular (e.g., closed) Piezo configuration (*Figure 1A and B*).

Since the surrounding lipid membrane connects smoothly to the Piezo dome, the curved shape of Piezo is expected to induce membrane curvature beyond the perimeter of the Piezo dome. The fundamental reason for this is, the energetic cost to curve a membrane contains a term proportional to the membrane's mean curvature squared. As a result, a sharp transition from the curved dome shape to a planar membrane is associated with a higher energy than a gradual transition. This effect is

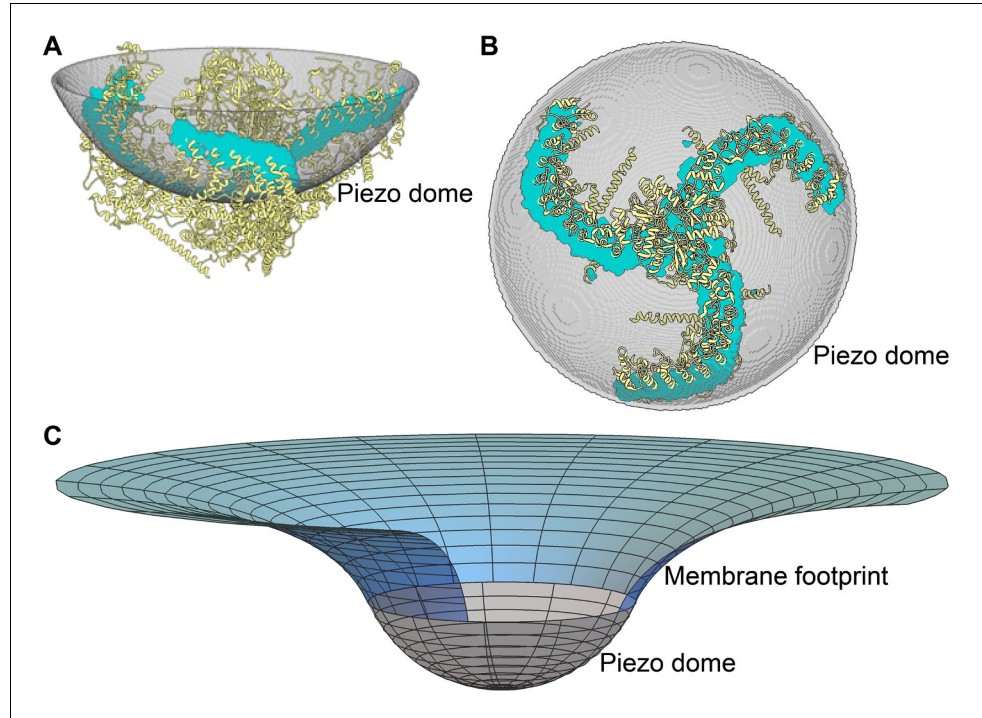

**Figure 1.** Piezo curves the membrane. (**A**) Side and (**B**) top-down (projecting into the cell) views of the Piezo dome. The approximate position of the curved mid-bilayer surface of the Piezo dome is indicated in grey, with the cyan regions corresponding to the intersection of the mid-bilayer surface and the Piezo protein. (**C**) The curved shape of the mid-bilayer surface of the Piezo dome (indicated in grey) deforms the mid-bilayer surface of the surrounding lipid membrane (indicated in blue) and results in a membrane footprint of Piezo that extends beyond the size of the dome (see **Figure 2A** for further details). [The atomic structure of the Piezo protein in (**A**) and (**B**) corresponds to mPiezo1 with Protein Data Bank (http://www.rcsb.org) ID 6B3R.].

DOI: https://doi.org/10.7554/eLife.41968.002

shown in **Figure 1C**: the grey surface corresponds to the mid-bilayer surface of the dome in **Figure 1A and B** and the blue surface to the mid-bilayer surface of the surrounding membrane. We refer to the region of deformed lipid membrane outside the perimeter of the Piezo dome as Piezo's membrane footprint (**Phillips et al., 2009**). The total energy of the membrane-Piezo system therefore has to include Piezo's membrane footprint in addition to the Piezo dome. As we will show, Piezo's membrane footprint not only influences the total energy of the membrane-Piezo system, but it also has a very large influence on Piezo's ability to sense changes in membrane tension.

## Shape and energy of the membrane footprint

Of all the possible shapes Piezo's membrane footprint may adopt, we assume that the dominant shape corresponds to that associated with the lowest energy. To calculate this lowest energy membrane footprint, we begin with a well-known expression for the lipid membrane deformation energy (**Helfrich, 1973**)

$$G_M = \frac{1}{2} K_b \int (c_1 + c_2)^2 dA + \gamma \, \Delta A \,, \qquad (1)$$

where $K_b$ is the membrane bending modulus (membrane bending stiffness), $\gamma$ is the membrane tension, $c_1$ and $c_2$ are the principal curvatures of the mid-bilayer surface (which are functions of position on the membrane), and $\Delta A$ is the decrease in in-plane area associated with deforming the membrane out of its unperturbed (planar) configuration. The integration is carried out over the surface of the membrane footprint (see Appendix 1). In the integrand of the membrane bending energy in **Equation 1** we did not include a contribution $\propto c_1 c_2$ due to the Gaussian curvature of the membrane,

which is independent of the shape of the membrane footprint, and a contribution due to the membrane spontaneous curvature (*Helfrich, 1973*). The latter contribution to the membrane bending energy may need to be considered if the bilayer contains lipids that induce intrinsic curvature.

Next, we minimize $G_M$ by solving a differential equation corresponding to the first variation of *Equation 1* set equal to zero – the Euler-Lagrange equation – subject to specific boundary conditions (*Fox, 1987*). This solution yields the shape of the lipid membrane when its energy is minimal. Using this shape, we calculate $G_M$ by evaluating *Equation 1*. We used two separate, previously developed methods – one analytical (*Weikl et al., 1998*; *Turner and Sens, 2004*; *Wiggins and Phillips, 2005*; *Li et al., 2017*) and one numerical (*Peterson, 1985*; *Seifert et al., 1991*; *Deserno, 2004*; *Bahrami et al., 2016*) – to carry out these calculations. The analytical solutions involve a 'small gradient' approximation of *Equation 1* and are therefore only accurate for cases in which the membrane curvature is small. Nevertheless, the analytical solutions provide an important check (see Materials and methods) on the numerical solutions, which are not limited to membranes with small curvatures. Because Piezo can be highly curved, the solutions shown in the main text figures were calculated numerically.

The shape of Piezo's membrane footprint – and therefore its associated energy – depends on three key physical properties of the membrane-Piezo system: the basic shape of the Piezo dome, the membrane bending modulus $K_b$, and the membrane tension $\gamma$. The general shape of Piezo in a closed conformation is well defined and approximated here as a dome, or spherical cap, of area $390 \text{ nm}^2$ and radius of curvature $R = R_c$ with $R_c = 10.2 \text{ nm}$ (*Guo and MacKinnon, 2017*). We assume that the area of the Piezo dome stays approximately constant independent of the conformational state of Piezo. The value of $K_b$ for membranes with lipid compositions common to cell membranes is well documented, around $20 \, k_B T$ (*Rawicz et al., 2000*), and values of $\gamma$ relevant to living cells and required to activate Piezo have been described (*Lewis and Grandl, 2015*; *Cox et al., 2016*). Therefore, calculation of Piezo's membrane footprint and its associated energy is a well-defined mechanics problem involving no free parameters.

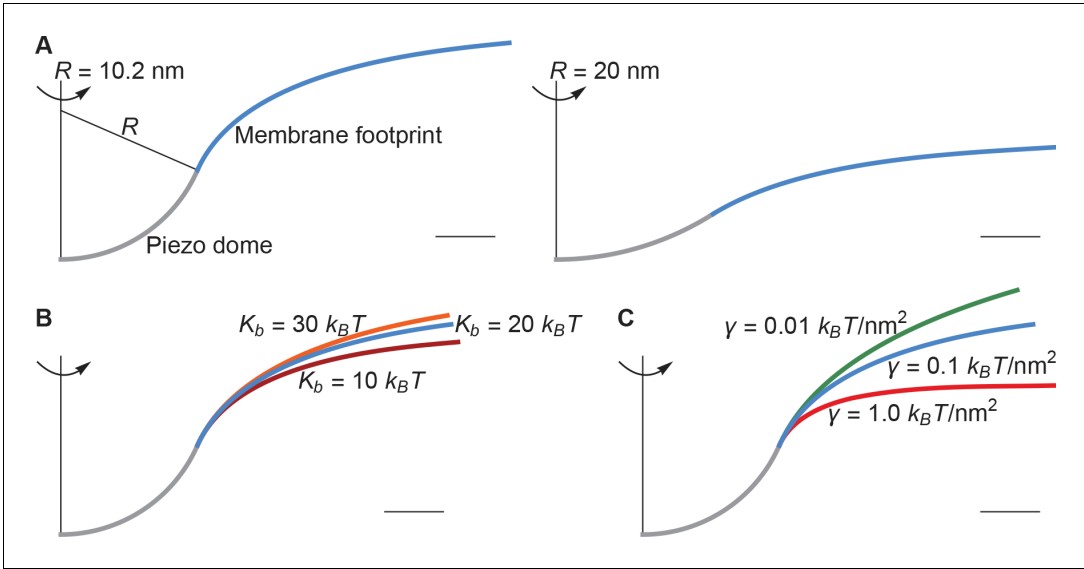

**Figure 2.** Membrane footprint of the Piezo dome. The shape of the Piezo membrane footprint depends on (**A**) the radius of curvature of the Piezo dome $R$, (**B**) the membrane bending modulus (membrane bending stiffness) $K_b$, and (**C**) the membrane tension $\gamma$. All curves show the cross section of the mid-bilayer surface and its intersection with the Piezo protein. Unless indicated otherwise, we calculated the Piezo membrane footprint using the value $R = 10.2 \text{ nm}$ observed for Piezo in a closed conformation (*Guo and MacKinnon, 2017*) with $K_b = 20 \, k_B T$ and $\gamma = 0.1 \, k_B T/\text{nm}^2$. For *Figure 1C* we used the same parameter values as in the left panel of *Figure 2A*. The range of $K_b$ considered in (**B**) corresponds to the approximate range of $K_b$ measured for phosphatidylcholine bilayers with different acyl-chain lengths and degrees of unsaturation (*Rawicz et al., 2000*). Scale bars, $4 \text{ nm}$.
DOI: https://doi.org/10.7554/eLife.41968.003

The left panel of *Figure 2A* shows a cross section through the surface displayed in *Figure 1C*, calculated as described above, corresponding to $R = 10.2$ nm, $K_b = 20\,k_BT$, and $\gamma = 0.1\,k_BT/\mathrm{nm}^2$ ($1\,k_BT/\mathrm{nm}^2 = 4.114$ mN/m at $T = 298$ K). For context on this value of the membrane tension, commonly studied membranes undergo lysis at around $3.5\,k_BT/\mathrm{nm}^2$ (*Rawicz et al., 2000*). Thus, $0.1\,k_BT/\mathrm{nm}^2$ is a modest value of the membrane tension, likely experienced by cell membranes under non-pathological stress. The left panel of *Figure 2A* illustrates that, if one includes the membrane footprint, then Piezo has an extensive reach and, as we will show, this reach has significant functional consequences. But first we inspect how the three physical properties $R$, $K_b$, and $\gamma$ of the membrane-Piezo system affect the size and shape of Piezo's membrane footprint. If $R$ were to be increased (i.e., if Piezo were to become flatter) then the deformation footprint would become less pronounced and smaller in height (right panel of *Figure 2A*). The magnitudes of $K_b$ and $\gamma$ change the reach of the membrane footprint: larger $K_b$ and smaller $\gamma$ values produce a more gradual approach to the plane of the membrane (*Figure 2B and C*). This relationship is expressed by the characteristic decay length of membrane shape deformations,

$$\lambda = \sqrt{K_b/\gamma},\tag{2}$$

which appears in the analytical solution to the Euler-Lagrange equation associated with *Equation 1* (Appendix 1, *Equations A6 and A7*). Substituting $K_b = 20\,k_BT$ and $\gamma = 0.1\,k_BT/\mathrm{nm}^2$ yields $\lambda = 14$ nm, which means that under these conditions Piezo's membrane footprint is much larger than the Piezo protein itself.

The membrane footprint energy, $G_M$, is graphed in *Figure 3* as a function of Piezo's radius of curvature. $G_M$ is greater than or equal to zero because this energy represents the work required to deform the membrane from a plane into the shape of Piezo's membrane footprint. *Figure 3A* shows the energetic consequence if Piezo could undergo a conformational transition that changes its radius of curvature: a highly curved Piezo (small $R$) is associated with a large $G_M$. We also see that $G_M$ is a sensitive function of membrane tension. If Piezo becomes flatter when it opens, as was proposed in the membrane dome mechanism (*Guo and MacKinnon, 2017*), then the deformation footprint will contribute to the energetics of gating, as shown (*Figure 3B*). We denote here the radii of curvature of the Piezo dome in the closed and open conformational states of Piezo by $R_c$ and $R_o$, with $R_c < R_o$. Under finite membrane tension ($\gamma > 0$) Piezo flattening (i.e., a transition from $R = R_c$ to $R = R_o$) will reduce $G_M$ and thus stabilize the flatter, open conformation relative to the closed conformation. In the absence of membrane tension ($\gamma = 0$) the membrane footprint is of no energetic consequence.

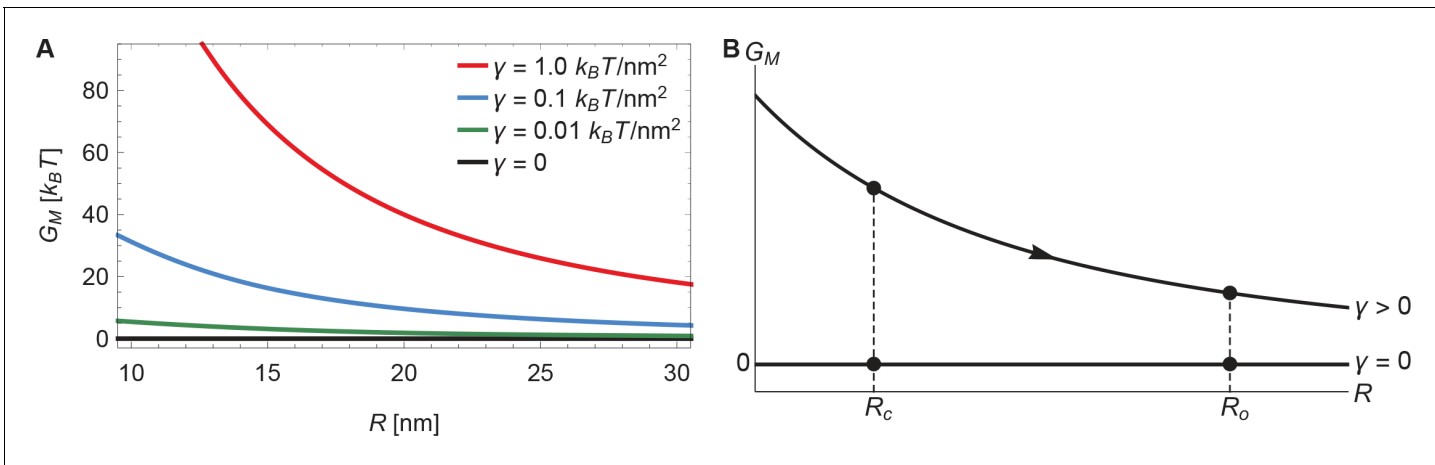

**Figure 3.** Energy of the Piezo membrane footprint. (A) Energy cost of the Piezo membrane footprint $G_M$ as a function of the radius of curvature of the Piezo dome $R$. We calculated $G_M$ by minimizing *Equation 1* with the membrane bending rigidity $K_b = 20\,k_BT$ and the indicated values of the membrane tension $\gamma$. (B) Schematic of the proposed mechanism for the mechanical activation of Piezo through membrane tension, for which we assume that the radius of curvature of the Piezo dome in the closed conformational state, $R_c$, takes a smaller value than in the open conformational state, $R_o$.
DOI: https://doi.org/10.7554/eLife.41968.004

Thus, Piezo's membrane footprint would impose a tension-dependent bias, favoring the open conformation of Piezo only when tension is applied, and more so when tension is greater.

Piezo's membrane footprint in the absence of applied tension, which is associated with $G_M = 0$, deserves a comment because the membrane is still highly curved here (see *Figure 2C* as $\gamma$ becomes smaller). If $G_M$ represents the work required to deform the membrane from a plane into the shape of Piezo's membrane footprint, and Piezo's membrane footprint is curved, how can $G_M$ be zero? The explanation is that, in the limit $\gamma \to 0$, the membrane curves in a special way around the Piezo dome such that the principal curvatures $c_1$ and $c_2$ in *Equation 1* sum to zero. This special surface, called a catenoid, would never truly be achieved in this physical system because thermal fluctuations will not permit zero tension and, potentially, because of deviations of the Piezo dome from a perfect spherical cap. Nevertheless, in the absence of applied tension the deformation footprint should approach the approximate shape of a catenoid. As we demonstrate below, this behavior yields fascinating consequences for Piezo's mechanosensitivity.

## Influence of the membrane footprint on gating

The above analysis suggests that $G_M$, the energy required to form Piezo's membrane footprint, should influence the gating properties of Piezo. To investigate the nature of this influence, we add to the Piezo dome energy the energetic contribution due to Piezo's membrane footprint. The dome energy, $G_D$, has three additive contributions (*Guo and MacKinnon, 2017*): the protein energy $G_D^P$, in which we include all contributions to the energy of the membrane-Piezo system that do not depend on the membrane tension or the membrane shape, the energy required to bend the membrane in between Piezo's arms (still part of the dome) against membrane bending stiffness, $G_D^b$, and the work required to form the dome against membrane tension, $G_D^\gamma$. The total energy of the membrane-Piezo system is therefore given by

$$G = G_D^P + G_D^b + G_D^\gamma + G_M. \qquad (3)$$

$G$ is the work required to form both the Piezo dome (i.e., the curved Piezo protein and the curved membrane between the arms) and Piezo's membrane footprint, starting from a hypothetically planar standard state. The value of $G_D^P$ is unknown, $G_D^b$ was estimated previously to be $2.4\,\pi\,K_b$ (approximating all of the dome area to be occupied by lipids), and $G_D^\gamma = \gamma\,\Delta A$ with, similarly as above, $\Delta A$ being the decrease in the in-plane area of the Piezo dome compared to the planar state (*Guo and MacKinnon, 2017*). In addition to internal protein interactions, $G_D^P$ may include a contribution to the membrane bending energy due to the Gaussian curvature of the membrane (*Helfrich, 1973*). The Gauss-Bonnet theorem mandates that, for a fixed membrane topology, this contribution to $G_D^P$ only depends on the boundaries of the membrane, and hence takes a constant value for a given Piezo conformational state and membrane composition (*Weikl et al., 1998*; *Wiggins and Phillips, 2005*).

Now, if the dome increases its radius of curvature when Piezo opens, then the total energy difference between the open and closed conformations, $\Delta G$, is obtained by applying *Equation 3* to each conformation and taking the difference. The upper panel of *Figure 4A* shows this difference for the tension-dependent components of $\Delta G$, $\Delta G_D^\gamma$ and $\Delta G_M$, for a closed to open transition if $R_c = 10.2$ nm and $R_o \to \infty$ (i.e., Piezo being flat in the open conformation), as a function of $\gamma$. $\Delta G_D^\gamma$ and $\Delta G_M$ are plotted separately for comparison. It is immediately clear that $\Delta G_M$ is expected to contribute substantially to Piezo's tension-dependent gating. Two other possible geometries, corresponding to a smaller degree of flattening (*Figure 4B*), or to flattening from a less curved closed state (*Figure 4C*), are also shown. We consider the former geometry to explore the decrease in Piezo curvature required for mechanosensitivity, and the latter geometry because the curvature of the Piezo dome may be reduced in cellular membrane environments. In all three cases, for tension values likely relevant to Piezo gating, the contribution to the Piezo gating energy due to Piezo's membrane footprint is approximately equal to or greater than the tension-dependent contribution due to the Piezo dome itself.

The tension sensitivity of Piezo gating depends on how steeply $\Delta G$ changes with respect to a change in $\gamma$, $d\Delta G/d\gamma$. We graph the predicted tension sensitivity of Piezo gating in the lower panels of *Figure 4A–C*, again with the contributions due to the Piezo dome and Piezo's membrane footprint separated for comparison. The negative sign indicates that increasing $\gamma$ favors the open conformation. For the dome, sensitivity is constant, equal to a constant change in $\Delta A$. For the membrane

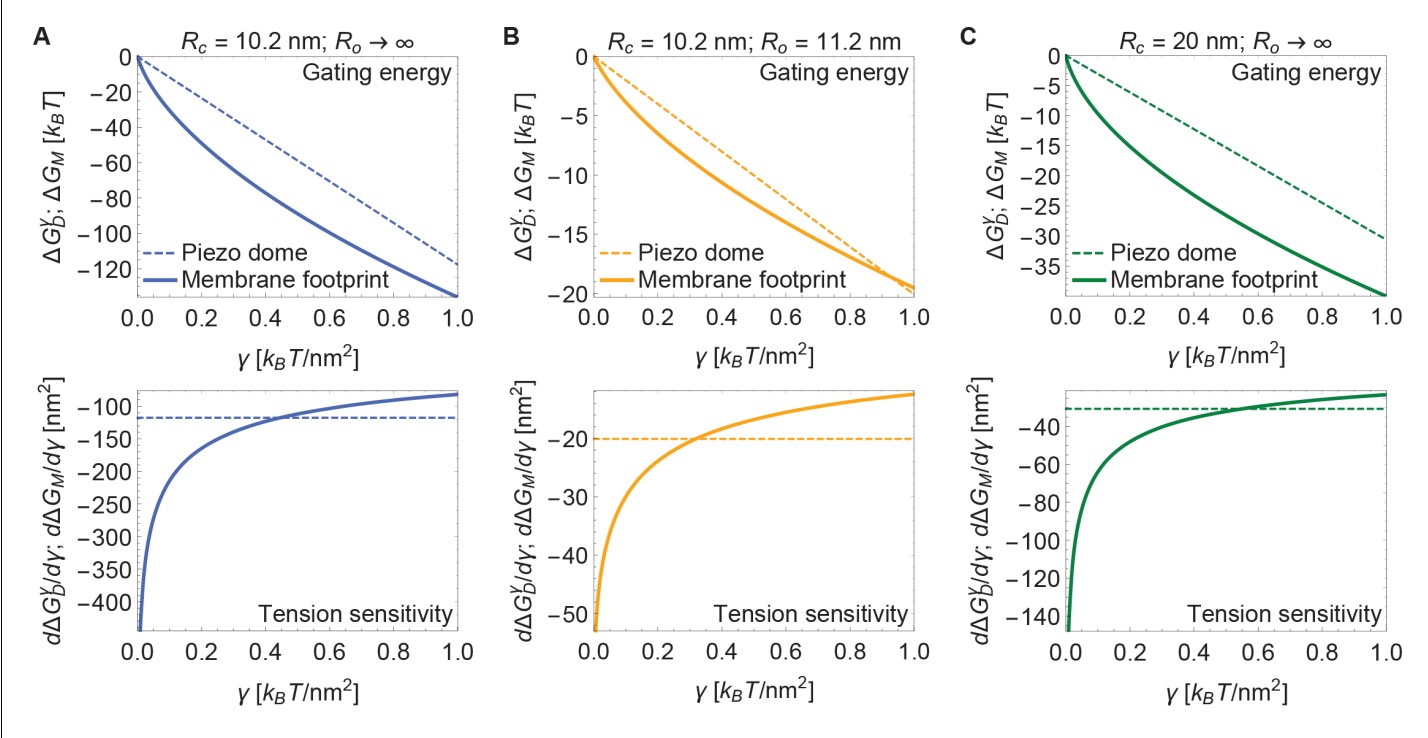

**Figure 4.** Energy of Piezo gating. Tension-dependent contributions to the Piezo gating energy (upper panels) and associated tension sensitivity (lower panels) due to the Piezo dome, $\Delta G_D^\gamma$, and the Piezo membrane footprint, $\Delta G_M$, as a function of membrane tension for the Piezo dome radii of curvature (A) $R_c = 10.2$ nm and $R_o \to \infty$, (B) $R_c = 10.2$ nm and $R_o = 11.2$ nm, and (C) $R_c = 20$ nm and $R_o \to \infty$ in the closed and open conformational states of Piezo, respectively. For all calculations, we set the membrane bending rigidity $K_b = 20$ $k_BT$.

DOI: https://doi.org/10.7554/eLife.41968.005

footprint, the magnitude of the sensitivity is not constant and very large for small $\gamma$. In fact, using the analytical approach for calculating Piezo's membrane footprint it can be shown that the tension sensitivity grows without bound as the membrane tension approaches zero. This remarkable result means that Piezo's membrane footprint renders Piezo exquisitely sensitive in the low-tension regime; most sensitive to the smallest perturbations around zero tension. The diverging tension sensitivity as $\gamma \to 0$ is a consequence of the idealized catenoidal membrane footprint that is formed at zero tension. The membrane footprint is large and curved, but in a special manner. Once an incrementally small value of membrane tension is applied, this large, previously energy-free, membrane footprint is available to release in-plane area and to unbend, reducing the free energy of the expanded (open) conformation relative to the closed conformation.

*Figure 5* presents open probability ($P_o$) and gating sensitivity ($dP_o/d\gamma$) curves for the energy values in *Figure 4*, applied to a 2-state gating model, for which

$$\frac{P_o}{1 - P_o} = e^{-\Delta G/k_B T}. \tag{4}$$

The unknown values of $G_D^P$ were chosen so that opening occurs within the tension range shown. Since $G_D^P$ is unknown, the Piezo gating tension is not a model prediction. The solid and dashed curves correspond to the gating response with and without inclusion of Piezo's membrane footprint energy. The membrane footprint energy shifts the $P_o$ curve in the direction of smaller tension values and steepens it (i.e., increases its sensitivity). The particular gating curves shown here depend on a specific, simple gating equilibrium scheme and an unknown value of $\Delta G_D^P$. Because the contribution of Piezo's membrane footprint to the Piezo gating energy is so large, however, the conclusion that the position and steepness of the $P_o$ curve should exhibit a strong dependence on Piezo's membrane footprint will apply to a wide range of possible gating schemes.

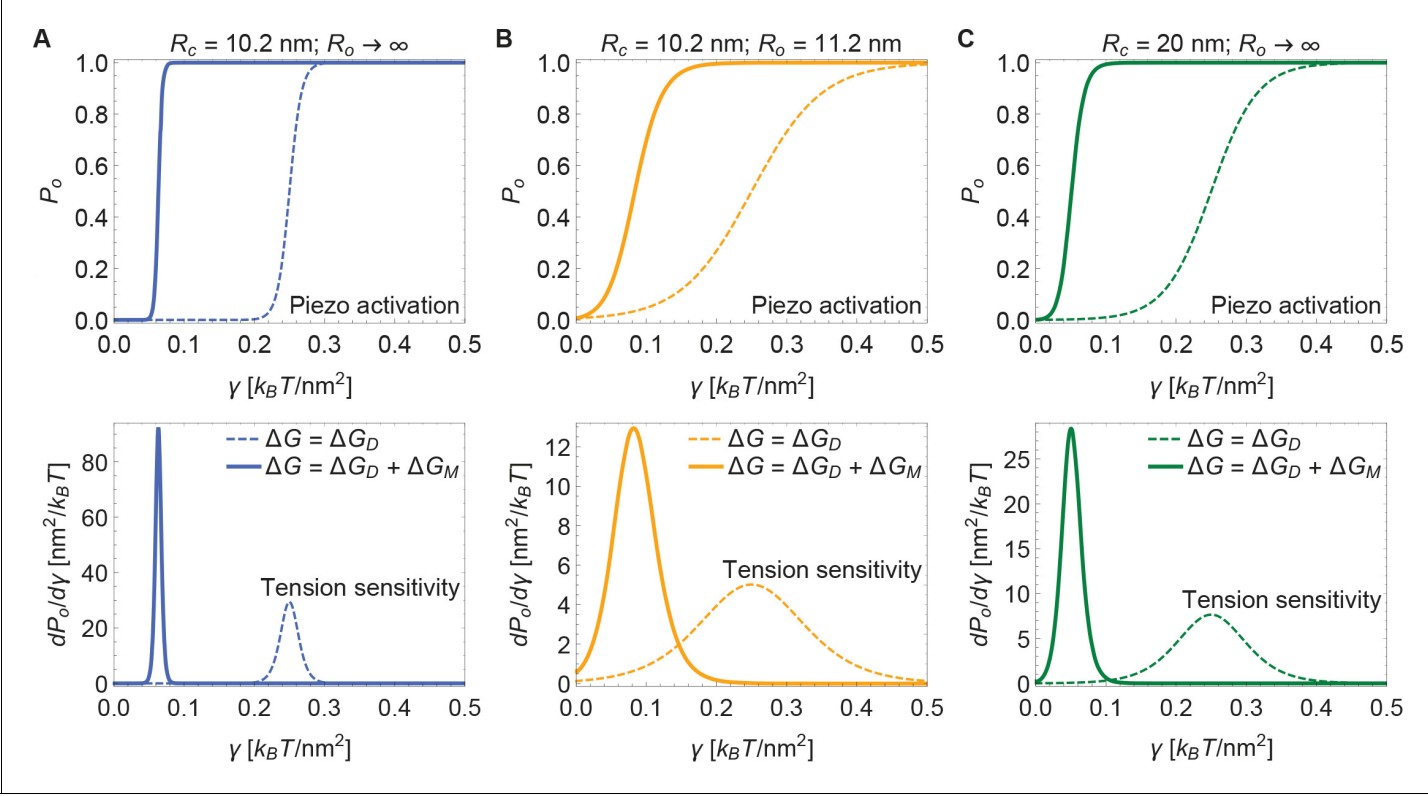

**Figure 5.** Piezo activation through membrane tension. Piezo activation curves $P_o$ (upper panels) and associated tension sensitivity (lower panels) resulting solely from the gating energy due to the Piezo dome, $\Delta G = \Delta G_D$, and from the gating energy due to the Piezo dome together with the Piezo membrane footprint, $\Delta G = \Delta G_D + \Delta G_M$, as a function of membrane tension for the Piezo dome radii of curvature (A) $R_c = 10.2$ nm and $R_o \to \infty$, (B) $R_c = 10.2$ nm and $R_o = 11.2$ nm, and (C) $R_c = 20$ nm and $R_o \to \infty$ in the closed and open conformational states of Piezo, respectively. For all calculations, we set the membrane bending rigidity $K_b = 20\ k_BT$. We used the values (A) $\Delta G_D^P \approx 180\ k_BT$, (B) $\Delta G_D^P \approx 31\ k_BT$, and (C) $\Delta G_D^P \approx 47\ k_BT$ for the (unknown) contribution of the protein energy to the Piezo gating energy such that gating occurs within the indicated tension range.

DOI: https://doi.org/10.7554/eLife.41968.006

## Modulation of gating through the membrane

Next, we consider the influence of membrane bending stiffness on Piezo gating. We quantify the magnitude of the membrane bending stiffness by the membrane bending modulus $K_b$. We are interested in this dependence because membrane bending stiffness is a function of lipid composition, which could vary among different cell types and possibly even between different regions within the same cell. To what extent might membrane bending stiffness influence Piezo's response to membrane tension? Membrane bending stiffness enters the Piezo gating energy through the dome contribution $\Delta G_D^b$ and the footprint contribution $\Delta G_M$. *Figure 6A* shows the sum of these two membrane bending stiffness-dependent contributions to the Piezo gating energy and associated $P_o$ and sensitivity curves for three different values of $K_b$. Note that $\Delta G_D^b$ contributes as a tension-independent constant, whereas the contribution $\Delta G_M$ depends on membrane tension. Together, $\Delta G_D^b$ and $\Delta G_M$ contribute significantly to $\Delta G$ and thus to gating. This implies that Piezo channels in different cell types and possibly different locations within a cell will exhibit different gating characteristics.

The membrane footprint induced by Piezo is expected to influence the distribution of molecules – both lipids and proteins – in the surrounding membrane. Piezo's membrane footprint should attract lipids and proteins that exhibit an energetic preference for the curved shape of the membrane footprint, and repel molecules that 'prefer' other membrane shapes. Conversely, our model of Piezo's membrane footprint implies that the composition of the surrounding membrane should influence the energetics of Piezo gating. This model prediction raises interesting possibilities for the regulation of Piezo gating in different membrane environments. The membrane footprint induced by

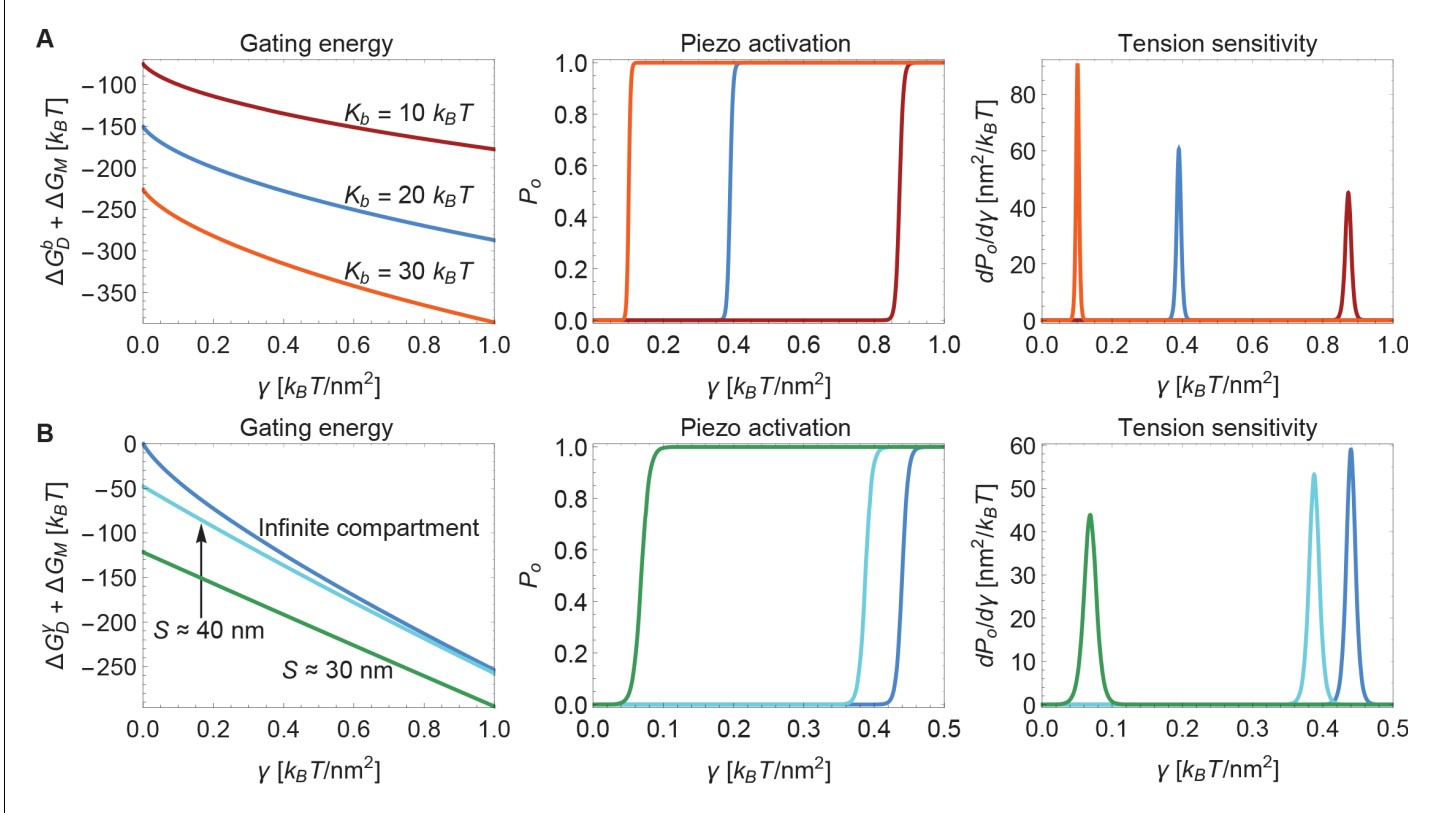

**Figure 6.** Modulation of Piezo gating through the membrane. (A) Membrane bending stiffness-dependent contribution to the Piezo gating energy $\Delta G_D^b + \Delta G_M$ (left panel) and associated Piezo activation and tension sensitivity curves (middle and right panels) as a function of membrane tension for the indicated values of the membrane bending stiffness $K_b$. (B) Membrane tension-dependent contribution to the Piezo gating energy $\Delta G_D^\gamma + \Delta G_M$ (left panel) and associated Piezo activation and tension sensitivity curves (middle and right panels) as a function of membrane tension for infinite and finite membrane compartments. For both (A) and (B) we employed the Piezo dome radii of curvature $R_c = 10.2$ nm and $R_o \to \infty$ in the closed and open conformational states of Piezo, respectively. For (B) we used the unconstrained membrane arc lengths 5 nm and 10 nm separating the border of the Piezo dome and the border of the membrane compartment along the membrane in the radial direction, which correspond to the membrane compartment diameters $S \approx 30$ nm and $S \approx 40$ nm, respectively, and set $K_b = 20$ $k_B T$. We calculated the curves in the middle and right panels of (A) and (B) from the total energy of the membrane-Piezo system in **Equation 3**, with the values (A) $\Delta G_D^P \approx 270$ $k_B T$ and (B) $\Delta G_D^P \approx 280$ $k_B T$ for the (unknown) contribution of the protein energy to the Piezo gating energy such that gating occurs within the indicated tension ranges.
DOI: https://doi.org/10.7554/eLife.41968.007

Piezo also implies that Piezo channels should interact with each other through the membrane, and hence influence each other's local distribution in the membrane and gating properties.

Finally, we consider the effect of membrane compartment size on Piezo gating. In the calculations presented so far Piezo was assumed to reside in an infinite membrane that approaches a planar configuration far from the channel. But real cell membranes are compartmentalized. For example, cytoskeletal attachments, which occur at spatial frequencies of up to tens of nanometers, can restrict the shapes a membrane can take (**Kusumi et al., 2014**). **Figure 6B** shows the sum of the tension-dependent contributions to the Piezo gating energy, $\Delta G_D^\gamma$ and $\Delta G_M$, and associated $P_o$ and sensitivity curves, for different compartmental restrictions. Membrane compartments with diameters $S$ approximately equal to 30 nm and 40 nm are compared to an infinite membrane. These compartments restrict the distance between Piezo's outer perimeter and the edge of the membrane compartment to distances of 5 nm and 10 nm along the membrane in the radial direction, respectively. In general, the effects of membrane compartmentalization are greater in the low-tension regime. This result can be understood in terms of the characteristic decay length of membrane shape deformations in **Equation 2**: larger values of $\gamma$ reduce the size of Piezo's membrane footprint so that it fits better into the membrane compartment.

We also note that the smaller the membrane compartment, the greater influence it has on Piezo gating. This is because in these particular calculations the membrane is constrained to planarity at the edge of the membrane compartment, but the effect will in general also depend on the membrane slope constraint at the edge of the membrane compartment. The important point is that membrane compartmentalization can have a large effect on Piezo gating because membrane compartments can alter the shape and therefore the energy of Piezo's membrane footprint. Experimentally observed effects of cytoskeletal removal on some properties of Piezo gating could reflect the importance of Piezo's membrane footprint for Piezo gating (*Cox et al., 2016*).

In *Figure 6* we neglected the contribution to the membrane bending energy due to the Gaussian curvature of the membrane (*Helfrich, 1973*). While being independent of the shape of the membrane footprint, the Gaussian contribution to the membrane bending energy depends on the membrane composition and on how the membrane is constrained at the Piezo and membrane compartment boundaries (*Weikl et al., 1998*; *Wiggins and Phillips, 2005*). Contributions to the membrane bending energy due to the Gaussian membrane curvature may therefore further modulate Piezo gating in compartmentalized membranes with heterogeneous lipid compositions.

## Discussion

While Piezo channels can exhibit complex gating properties, including inactivation and voltage dependence, their dominant functional characteristic is that they open in response to mechanical force (*Coste et al., 2010*; *Lewis and Grandl, 2015*). This paper analyzes the influence of Piezo's unusual dome shape on the lipid bilayer membrane that surrounds the channel. The results depend on three key properties of the membrane-Piezo system and they are known: Piezo's shape, the lipid bilayer bending modulus, and the levels of tension that can be applied to a lipid membrane. Finding the shape of the lipid membrane surrounding Piezo, and its associated energy, amounts to solving a simple mechanics problem. And the inescapable conclusion is that Piezo, owing to its unusual shape, imposes a large structural perturbation – a deformation called a membrane footprint – on its surrounding membrane.

Depending on the applied membrane tension, Piezo's membrane footprint can come with a large energetic cost. Consequently, if Piezo changes its shape, for example if it becomes flatter upon opening, then the surrounding membrane will weigh in prominently in an energetic sense to Piezo's tension sensitivity. Moreover, Piezo's membrane footprint weighs in in such a way that the tension sensitivity of Piezo gating is greatest in the low-tension regime. This property would seem to render Piezo poised to respond to the slightest changes in cell membrane tension.

In our analysis of Piezo's membrane footprint we used a spherical dome shape to approximate a more complex underlying geometry of Piezo. Deviations from a spherical dome shape will alter the shape and energetics of the membrane footprint. But the basic idea that Piezo's curved shape will create an energetically important membrane footprint will still apply.

Piezo's large membrane footprint rationalizes what at first seemed to be a contradiction in the experimental literature. Certain data show the clear importance of the membrane in mediating Piezo's mechanosensitivity (*Lewis and Grandl, 2015*; *Syeda et al., 2016*; *Cox et al., 2016*), while other data show the importance of tether attachments (e.g., the cytoskeleton) to the channel or the membrane (*Cox et al., 2016*; *Wu et al., 2016*). A large membrane footprint essentially demands that both contributions be energetically important.

Piezo is a very uniquely shaped membrane protein. We think that this shape evolved specifically to exploit the physical properties of the lipid membrane to create a large membrane footprint, enabling exquisite tension sensitivity.

## Materials and methods

We used *Equation 1* to analytically and numerically calculate Piezo's membrane footprint, and its associated energy, through the Monge (*Weikl et al., 1998*; *Turner and Sens, 2004*; *Wiggins and Phillips, 2005*; *Li et al., 2017*) and arclength (*Peterson, 1985*; *Seifert et al., 1991*; *Deserno, 2004*; *Bahrami et al., 2016*) parametrizations of surfaces, respectively. Appendix 1-sections 1 and 2 provide a detailed discussion of these Monge and arclength solutions. All of the results shown in the main text figures were calculated numerically using the arclength parametrization of surfaces, which

allows for large membrane curvatures. In Appendix 1-section 3 we compare the analytical and numerical solutions obtained using the Monge and arclength parametrizations of surfaces. We find that the Monge parametrization of surfaces tends to overestimate the magnitudes of Piezo's membrane footprint and its associated membrane deformation energy but yields, for the scenarios considered in the main text figures, qualitatively similar predictions as the arclength parameterization of surfaces.

## Acknowledgements

We thank Yusong R Guo for preparing *Figure 1A and B*, and Osman Kahraman and Rob Phillips for helpful discussions. RM is an Investigator in the Howard Hughes Medical Institute.

## Additional information

### Funding

| Funder | Grant reference number | Author |
| --- | --- | --- |
| National Science Foundation | DMR-1554716 | Christoph A Haselwandter |
| Howard Hughes Medical Institute | | Roderick MacKinnon |

The funders had no role in study design, data collection and interpretation, or the decision to submit the work for publication.

### Author contributions
Christoph A Haselwandter, Roderick MacKinnon, Conceptualization, Formal analysis, Funding acquisition, Investigation, Visualization, Methodology, Writing—original draft, Writing—review and editing

### Author ORCIDs
Christoph A Haselwandter (iD) http://orcid.org/0000-0002-5012-5640
Roderick MacKinnon (iD) https://orcid.org/0000-0001-7605-4679

### Decision letter and Author response
Decision letter https://doi.org/10.7554/eLife.41968.018
Author response https://doi.org/10.7554/eLife.41968.019

## Additional files

### Supplementary files
• Transparent reporting form
DOI: https://doi.org/10.7554/eLife.41968.008

### Data availability
All data generated or analysed in this study are available through this manuscript.

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

## Appendix 1

DOI: https://doi.org/10.7554/eLife.41968.009

We provide here a detailed description of the mathematical methods we used for the calculation of the membrane footprint (see *Appendix 1—figure 1*), and associated membrane deformation energy, of the Piezo dome. As explained in the main text, we calculated the membrane footprint of the Piezo dome by finding the stationary membrane shapes associated with the classic Helfrich energy.

$$G_M = \frac{1}{2}K_b \int dA (c_1 + c_2)^2 + \gamma \Delta A \,, \qquad (A1)$$

where the integral runs over the membrane surface surrounding the Piezo dome, $c_1$ and $c_2$ are the local principal curvatures of the membrane surface, $K_b$ is the lipid bilayer bending modulus, $\gamma$ is the membrane tension, and $\Delta A$ is the decrease in in-plane area due to the membrane shape deformations induced by the Piezo dome (see also *Equation 1* of the main text). (A given stationary membrane shape may, in principle, be unstable to small perturbations, and hence may not be physically relevant (*Peterson, 1985*; *Seifert et al., 1991*; *Seifert, 1997*). Here, we only found one stationary membrane shape for each scenario considered, and therefore identified this stationary membrane shape with Piezo's membrane footprint. The functions which make a given functional such as in *Equation A1* stationary are referred to as the extremal functions of this functional (*Courant and Hilbert, 1953*).) For a given radius of curvature of the Piezo dome, $R$, we obtained the energy cost of Piezo's membrane footprint by substituting the corresponding stationary membrane shape implied by *Equation A1* back into *Equation A1* and evaluating the surface integral.

We obtained the stationary membrane shapes associated with *Equation A1*, and calculated the corresponding energy cost of Piezo's membrane footprint, using two complementary mathematical approaches. On the one hand, we used the Monge parametrization of surfaces to derive exact analytical solutions of the stationary Piezo membrane footprints implied by *Equation A1*, and then used *Equation A1* to determine the corresponding exact analytic expressions for $G_M$ (*Weikl et al., 1998*; *Turner and Sens, 2004*; *Wiggins and Phillips, 2005*; *Ursell et al., 2008*; *Auth and Gompper, 2009*; *Sabass and Stone, 2016*; *Li et al., 2017*) (see Appendix 1-section 1). The Monge parametrization of surfaces is only expected to yield quantitatively accurate results for the stationary membrane shapes implied by *Equation A1* in the limit of small membrane shape deformations. We therefore used, on the other hand, the arclength parametrization of surfaces, which can capture arbitrarily large membrane shape deformations, to numerically solve for the membrane footprint of the Piezo dome and the corresponding $G_M$ (*Peterson, 1985*; *Seifert et al., 1991*; *Jülicher and Seifert, 1994*; *Deserno, 2004*; *Bahrami et al., 2016*) (see Appendix 1-section 2). Our calculations show that, for the radius of curvature $R = 10.2\,\mathrm{nm}$ measured previously for a closed state of Piezo (*Guo and MacKinnon, 2017*), the Piezo-induced membrane shape deformations are pronounced enough so that the Monge parametrization of *Equation A1* does not yield quantitatively accurate results. We find, however, that on a qualitative level the Monge and arclength parametrizations of *Equation A1* yield similar predictions for the mechanical gating of Piezo (see Appendix 1-section 3). All of the results shown in the main text were obtained using the arclength parametrization of *Equation A1*.

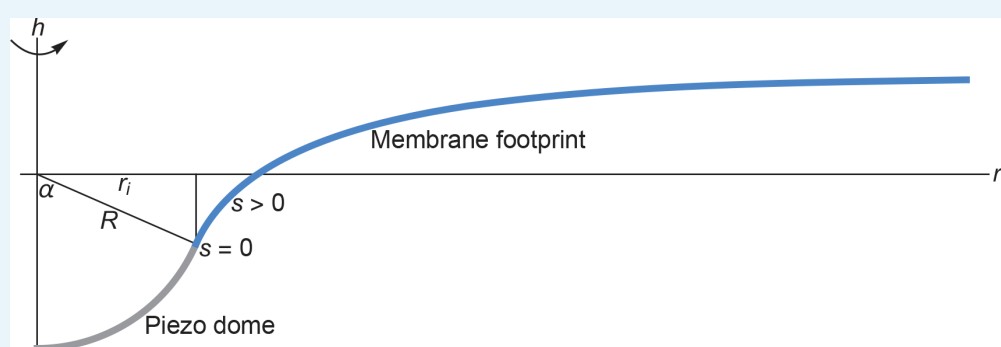

**Appendix 1—figure 1.** Cross section of the membrane shape deformations induced by Piezo for $K_b = 20\,k_BT$ and $\gamma = 0.1\,k_BT/\mathrm{nm}^2$. Based on the structural data in (**Guo and MacKinnon, 2017**) we assume that the Piezo dome takes the shape of a spherical cap with fixed cap area $S_{\mathrm{cap}} = 0.3 \times 4\pi \times 10.2^2\,\mathrm{nm}^2 \approx 390\,\mathrm{nm}^2$. We denote the radius of curvature of the Piezo dome by $R$, and represent the central pore axis of Piezo by the $h$-axis and the radial coordinate in the reference plane by $r$. Furthermore, we denote the arclength along the profile of Piezo's membrane footprint by $s$, with $s = 0$ at the interface of the Piezo dome and the surrounding membrane and $s > 0$ away from the Piezo dome. We set $R = 10.2\,\mathrm{nm}$ (**Guo and MacKinnon, 2017**) here and obtained Piezo's membrane footprint by numerically calculating the stationary membrane shape implied by the arclength parametrization of **Equation A1** (**Peterson, 1985**; **Seifert et al., 1991**; **Jülicher and Seifert, 1994**; **Deserno, 2004**; **Bahrami et al., 2016**) (see Appendix 1-section 2). The quantities $R$ and $S_{\mathrm{cap}}$ mathematically determine (**Weisstein, 2017**) the in-plane radius of the Piezo dome, $r = r_i$, and the cap angle, $\alpha$. We allowed here the Piezo-induced membrane shape deformations to decay to a flat membrane shape over an arbitrarily large $s$.

DOI: https://doi.org/10.7554/eLife.41968.010

## 1 Monge parametrization

For membrane profiles with no overhangs, Piezo's membrane footprint can be represented by the height of the lipid bilayer midplane $h(r)$ as a function of the radial coordinate $r$, with $r = 0$ corresponding to the central pore axis of Piezo, relative to some reference plane (see **Appendix 1—figure 1**). A particularly simple form of the energy in **Equation A1** is then obtained by assuming small membrane shape deformations in the membrane region surrounding Piezo, $|\nabla h| \ll 1$. To leading order in $|\nabla h|$ and its derivatives, the resulting Monge parametrization of **Equation A1** takes the form (**Weikl et al., 1998**; **Turner and Sens, 2004**; **Wiggins and Phillips, 2005**; **Ursell et al., 2008**; **Auth and Gompper, 2009**; **Sabass and Stone, 2016**; **Li et al., 2017**)

$$G = 2\pi \int dr\, r \left[ \frac{K_b}{2} \left( \nabla^2 h \right)^2 + \frac{\gamma}{2} \left( \nabla h \right)^2 \right], \tag{A2}$$

where, for simplicity, we have dropped the subscript '$M$' in **Equation A1**, and we have neglected terms that are constant in $h$ and its derivatives. Note that the energy in **Equation A2** only depends on $h$ through derivatives of $h$, and is therefore invariant under $h \mapsto h + h_0$, where $h_0$ is an arbitrary constant. For systems with rotational symmetry we have the Laplacian operator

$$\nabla^2 = \frac{d^2}{dr^2} + \frac{1}{r}\frac{d}{dr} \tag{A3}$$

in polar coordinates. The Piezo membrane deformation profile $h(r)$ is obtained by finding, subject to suitable boundary conditions on $h(r)$, the stationary membrane profile implied by **Equation A2**, from which the corresponding membrane deformation energy can be calculated

via *Equation A2*. The Monge parametrization of *Equation A1*, expressed in *Equation A2*, has the advantage that it provides simple analytic expressions for the membrane deformation energy associated with arbitrary conformational states of Piezo.

## 1.1 Membrane deformation profile

In our simple model of lipid bilayer-Piezo interactions we assume rotational symmetry about the central pore axis of Piezo with, potentially, a finite membrane compartment size (see the main text). The most general boundary value problem to be considered for *Equation A2* then corresponds to the membrane deformations in an annulus $r_i \leq r \leq r_m$, where $r_i$ and $r_m$ correspond to the radii of the inner and outer rims of the annulus. The value of $r_i$ is fixed (*Weisstein, 2017*) by the radius of curvature, $R$, and surface area, $S_{\text{cap}}$, associated with the Piezo dome (*Guo and MacKinnon, 2017*) (*Appendix 1—figure 1*),

$$r_i = \left[ \frac{S_{\text{cap}}}{4\pi^2 R^2} \left( 4\pi R^2 - S_{\text{cap}} \right) \right]^{1/2} . \tag{A4}$$

From *Equation A4* one finds (*Weisstein, 2017*) that the angle subtended by the $r$-axis in *Appendix 1—figure 1* and the tangent to $h(r)$ at the boundary between the Piezo dome and the surrounding membrane is given by

$$\alpha = \tan^{-1} \left\{ \frac{1}{2\pi R^2 - S_{\text{cap}}} \left[ S_{\text{cap}} \left( 4\pi R^2 - S_{\text{cap}} \right) \right]^{1/2} \right\} . \tag{A5}$$

We note that, as far as the membrane footprint of the Piezo dome is concerned, Piezo can effectively be regarded as a conical membrane inclusion of radius $r_i$ and opening angle $2\alpha$ (*Weikl et al., 1998*; *Turner and Sens, 2004*; *Wiggins and Phillips, 2005*; *Ursell et al., 2008*; *Auth and Gompper, 2009*; *Sabass and Stone, 2016*; *Li et al., 2017*).

The Euler-Lagrange (Lagrange) equation (*Courant and Hilbert, 1953*; *Kibble and Berkshire, 2004*) associated with *Equation A2* is given by (*Weikl et al., 1998*)

$$\nabla^2 \left( \nabla^2 - \lambda^{-2} \right) h = 0 , \tag{A6}$$

where $\lambda = \sqrt{K_b/\gamma}$ is the characteristic decay length of membrane shape deformations. For rotationally symmetric systems with the Laplacian operator in *Equation A3*, *Equation A6* has the general analytical solution (*Weikl et al., 1998*)

$$h(r) = A_0 I_0(r/\lambda) + B_0 K_0(r/\lambda) + C_0 + D_0 \ln r , \tag{A7}$$

where $I_0$ and $K_0$ are the zeroth-order modified Bessel functions of the first and second kind, and the constants $A_0$, $B_0$, $C_0$, and $D_0$ must be fixed through the boundary conditions on $h(r)$ at $r = r_i$ and/or $r = r_m$.

## 1.2 Boundary conditions

Consistent with previous work (*Weikl et al., 1998*; *Deserno, 2004*; *Turner and Sens, 2004*; *Wiggins and Phillips, 2005*; *Ursell et al., 2008*; *Auth and Gompper, 2009*; *Sabass and Stone, 2016*; *Li et al., 2017*), we demand continuity of $h(r)$ and its derivative at the boundary of the Piezo dome and the surrounding membrane. In particular, we fix the (arbitrary) height of the membrane-dome boundary relative to the reference plane,

$$h(r_i) = 0 , \tag{A8}$$

and impose

$$\left. \frac{dh}{dr} \right|_{r=r_i} = \tan \alpha \equiv a , \tag{A9}$$

where $\alpha$ is given by **Equation A5**. Having fixed $h(r_i) = 0$ via **Equation A8**, we take $h(r)$ to be free at the outer (membrane compartment) boundary $r = r_m$ (**Courant and Hilbert, 1953**). In other words, we assume that the difference of the membrane heights at the outer and inner boundaries, $h(r_m) - h(r_i)$, can be freely varied when finding the stationary membrane shape implied by **Equation A2**, resulting in the boundary condition (**Weikl et al., 1998**; **Auth and Gompper, 2009**; **Sabass and Stone, 2016**; **Li et al., 2017**)

$$\frac{d}{dr}\left[\nabla^2 h - \lambda^{-2} h\right]\bigg|_{r=r_m} = 0. \tag{A10}$$

**Equation A10** is sometimes referred to as a 'zero vertical force boundary condition' or a 'natural boundary condition' (**Courant and Hilbert, 1953**). Furthermore we take, in analogy to **Equation A9**, the membrane gradient to be fixed at the outer boundary (**Auth and Gompper, 2009**; **Li et al., 2017**),

$$\frac{dh}{dr}\bigg|_{r=r_m} = \tan\beta \equiv b, \tag{A11}$$

where we use here the value $b = 0$ corresponding to a flat membrane at the membrane compartment boundary. The boundary condition in **Equation A11** mathematically expresses the assumption that the membrane compartment boundary imposes a given (flat) membrane shape far enough away from Piezo.

The four boundary conditions in **Equations A8–A11** determine the four independent constants $A_0$, $B_0$, $C_0$, and $D_0$ in **Equation A7** (**Li et al., 2017**):

$$A_0 = \frac{bK_1(r_i/\lambda) - aK_1(r_m/\lambda)}{F}, \tag{A12}$$

$$B_0 = \frac{bI_1(r_i/\lambda) - aI_1(r_m/\lambda)}{F}, \tag{A13}$$

$$C_0 = \frac{aK_0(r_i/\lambda)I_1(r_m/\lambda) + aI_0(r_i/\lambda)K_1(r_m/\lambda) - b\lambda/r_i}{F}, \tag{A14}$$

$$D_0 = 0, \tag{A15}$$

where $I_1$ and $K_1$ are the first-order modified Bessel functions of the first and second kind, and

$$F = \frac{1}{\lambda}\left[K_1(r_i/\lambda)I_1(r_m/\lambda) - I_1(r_i/\lambda)K_1(r_m/\lambda)\right]. \tag{A16}$$

If, instead of **Equations A8 and A10**, we had imposed the boundary conditions

$$h(r_m) = 0, \tag{A17}$$

$$\frac{d}{dr}\left[\nabla^2 h - \lambda^{-2} h\right]\bigg|_{r=r_i} = 0, \tag{A18}$$

we would have obtained identical expressions for $A_0$, $B_0$, and $D_0$ in **Equations A12–A15**, but $C_0$ would be shifted so as to account for this redefinition of the (arbitrary) height of the membrane surface relative to the reference plane.

## 1.3 Analytic membrane deformation energy

Upon substitution of **Equation A7** into **Equation A2** and application of Gauss's theorem one obtains (**Li et al., 2017**)

$$G = \pi\gamma\{br_m[A_0 I_0(r_m/\lambda) + B_0 K_0(r_m/\lambda)] - ar_i[A_0 I_0(r_i/\lambda) + B_0 K_0(r_i/\lambda)]\}, \tag{A19}$$

where $A_0$ and $B_0$ are given by **Equations A12 and A13**. **Equation A19** is, within the Monge parametrization of **Equation A1**, expressed in **Equation A2**, the exact analytical solution of the energy cost associated with Piezo's membrane footprint. For $b = 0$, **Equation A19** reduces to

$$G = \pi K_b a^2 \frac{r_i}{\lambda} \frac{K_1(r_m/\lambda) I_0(r_i/\lambda) + I_1(r_m/\lambda) K_0(r_i/\lambda)}{K_1(r_i/\lambda) I_1(r_m/\lambda) - I_1(r_i/\lambda) K_1(r_m/\lambda)}. \tag{A20}$$

Note that *Equations A19 and A20* only depend on the particular solutions in *Equation A7* containing Bessel functions. These solutions are independent of the coefficient $C_0$ in *Equation A7*, and we would have obtained expressions identical to those in *Equations A19 and A20* if, instead of *Equations A8 and A10*, we had imposed the boundary conditions in *Equations A17 and A18*. This result can be understood by noting that the energy in *Equation A2* only depends on $h$ through derivatives of $h$, and hence does not depend on the absolute value of $h$.

The special case of infinitely large, asymptotically flat membranes (*Weikl et al., 1998*; *Deserno, 2004*; *Turner and Sens, 2004*; *Wiggins and Phillips, 2005*; *Ursell et al., 2008*; *Sabass and Stone, 2016*) corresponds to $b = 0$ and $r_m \to \infty$ in *Equations A7 and A19*. In this limit we have $I_1(r_m/\lambda) \to \infty$ and $K_1(r_m/\lambda) \to 0$. Via *Equation A20*, *Equation A19* then yields

$$G = \pi K_b a^2 \frac{r_i}{\lambda} \frac{K_0(r_i/\lambda)}{K_1(r_i/\lambda)}, \tag{A21}$$

and *Equation A7* with *Equations A12–A15* reduces to

$$h(r) = C_0 - a\lambda \frac{K_0(r/\lambda)}{K_1(r_i/\lambda)}. \tag{A22}$$

*Equations A21 and A22* agree with previous results (*Deserno, 2004*; *Wiggins and Phillips, 2005*) on the protein-induced lipid bilayer midplane deformations implied by the stationary shapes of *Equation A2* in asymptotically flat membranes. The limit $\gamma \to 0$ in *Equations A21 and A22* corresponds to $\lambda \to \infty$. Note that we have

$$K_0(x) \approx -\ln x, \tag{A23}$$

$$K_1(x) \approx \frac{1}{x} \tag{A24}$$

for $x \ll 1$. As $\lambda \to \infty$, *Equation A21* therefore yields

$$G \sim \frac{1}{\lambda^2} \ln(r_i/\lambda) \to 0, \tag{A25}$$

and *Equation A22* yields

$$h(r) \approx C_0 + a r_i \ln(r_i/\lambda). \tag{A26}$$

In agreement with previous studies (*Deserno, 2004*; *Auth and Gompper, 2009*), *Equations A25 and A26* show that, for a membrane that is asymptotically flat, the energy cost of Piezo's membrane footprint vanishes for $\gamma = 0$, with the membrane taking the shape of a minimal (catenoidal) surface. While this result followed from the Monge parametrization of *Equation A1* (i.e., in the limit of small membrane shape deformations), the arclength parametrization of *Equation A1* (see Appendix 1-section 2), which is valid for arbitrarily large membrane shape deformations, also yields for $\gamma = 0$ and asymptotically flat lipid membranes minimal (catenoidal) lipid bilayer deformations with zero bending energy (*Deserno, 2004*).

## 2 Arclength parametrization

The arclength parametrization of surfaces provides an elegant approach for the representation of axisymmetric shapes with, potentially, large gradients, and thus complements the Monge parametrization considered in Appendix 1-section 1. In the arclength parametrization, the Euler-Lagrange (Lagrange) equations associated with *Equation A1* are, in general, highly nonlinear (*Peterson, 1985*; *Seifert et al., 1991*; *Jülicher and Seifert, 1994*) and must be solved numerically. The arclength parametrization of lipid membrane surfaces has allowed the systematic determination of the minimum energy shapes of axisymmetric lipid bilayer vesicles (*Peterson, 1985*; *Seifert et al., 1991*; *Jülicher and Seifert, 1994*; *Seifert, 1997*), and has

been used to study the wrapping of spherical and cylindrical colloids by lipid bilayers (*Deserno and Bickel, 2003*; *Deserno, 2004*; *Hashemi et al., 2014*), endocytosis (*Nowak and Chou, 2008*; *Bahrami et al., 2016*; *Agudo-Canalejo and Lipowsky, 2016*), and the self-assembly of protein coats on lipid bilayer membranes (*Zhang and Nguyen, 2008*; *Foret, 2014*).

The arclength parametrization of *Equation A1* (*Peterson, 1985*; *Seifert et al., 1991*; *Jülicher and Seifert, 1994*) specifies the surface shape as a function of the arclength $s$ along the contour of the surface profile and the azimuthal angle about the symmetry axis of the system under consideration which, in the case of Piezo, corresponds to the central pore axis of Piezo. We denote the coordinate parallel to the axis of symmetry by $h(s)$, the (in-plane) coordinate perpendicular to the axis of symmetry by $r(s)$, and the angle between the tangent to the membrane deformation profile and the $r$-axis by $\psi(s)$ (*Appendix 1—figure 1*). (For the most part, we use here the same notation as *Deserno and Bickel, 2003* and *Deserno, 2004* for the arclength parametrization of *Equation A1*.) Note that $r(s)$ and $h(s)$ are geometrically related to $\psi(s)$ via

$$\dot{r} = \cos\psi, \tag{A27}$$

$$\dot{h} = \sin\psi, \tag{A28}$$

where we use the notation $\dot{r} \equiv dr/ds$ and $\dot{h} \equiv dh/ds$ in anticipation of $s$ being analogous to the 'time' coordinate in classical dynamics. In the arclength parametrization of surfaces, the energy in *Equation A1* takes the form (*Peterson, 1985*; *Seifert et al., 1991*; *Jülicher and Seifert, 1994*; *Deserno and Bickel, 2003*; *Deserno, 2004*; *Foret, 2014*)

$$G = \pi K_b \int_0^{s_0} ds\, r\left[\left(\dot{\psi} + \frac{\sin\psi}{r}\right)^2 + \frac{2}{\lambda^2}(1 - \cos\psi)\right] \tag{A29}$$

subject to the geometric constraints in *Equations A27 and A28*, where, for simplicity, we have dropped the subscript '$M$' in *Equation A1*, we have set $s = 0$ at the boundary of the Piezo dome and the surrounding membrane, $s > 0$ away from the Piezo dome, and $s_0 \to \infty$ for an infinite membrane. The terms $\dot{\psi}$ and $\sin\psi/r$ in *Equation A29* are the two principal curvatures of the membrane in the arclength parametrization (the term $\sin\psi/r$ can be rationalized by noting that the planes associated with the two principal curvatures must be perpendicular to each other, and the radius of curvature $= r/\sin\psi$ in the plane perpendicular to the $r$-$h$ plane in *Appendix 1—figure 1*), while the term $r\cos\psi$ yields the in-plane area of Piezo's membrane footprint (this can be seen by noting from *Equation A27* that $\cos\psi = dr/ds$), with the undeformed reference state of the membrane corresponding to a flat membrane with $\psi = 0$.

## 2.1 Hamilton equations

Incorporating the subsidiary conditions in *Equations A27 and A28* (*Courant and Hilbert, 1953*), the energy in *Equation A29* can be expressed as

$$G = \pi K_b \int_0^{s_0} ds L(\psi, \dot{\psi}, r, \dot{r}, \dot{h}), \tag{A30}$$

where

$$L = r\left[\left(\dot{\psi} + \frac{\sin\psi}{r}\right)^2 + \frac{2}{\lambda^2}(1 - \cos\psi)\right] + \lambda_r(\dot{r} - \cos\psi) + \lambda_h(\dot{h} - \sin\psi), \tag{A31}$$

in which the Lagrange parameter functions $\lambda_r(s)$ and $\lambda_h(s)$ must be chosen such that the constraints in *Equations A27 and A28* are satisfied by the extremal functions associated with *Equation A30*. The integrand $L$ in *Equation A30* is analogous to the Lagrangian function in classical dynamics (*Kibble and Berkshire, 2004*), with the arclength $s$ being the analogue of the time coordinate in classical dynamics.

The stationary membrane shapes implied by **Equation A30** could be obtained by directly solving the corresponding Euler-Lagrange equations (**Courant and Hilbert, 1953**; **Kibble and Berkshire, 2004**) subject to suitable boundary conditions on Piezo's membrane footprint. In particular, **Equation A30** is a function of the three generalized coordinates $q_{\psi,r,h} \equiv \psi, r, h$ capturing the shape of Piezo's membrane footprint, which implies that the corresponding Euler-Lagrange equations are given by a set of three coupled ordinary differential equations (**Courant and Hilbert, 1953**; **Kibble and Berkshire, 2004**),

$$\dot{p}_\delta = \frac{\partial L}{\partial q_\delta} \tag{A32}$$

for $\delta = \psi, r, h$, where the generalized momenta $p_\delta$ are defined by

$$p_\psi \equiv \frac{\partial L}{\partial \dot{q}_\psi} \equiv \frac{\partial L}{\partial \dot{\psi}} = 2r \left( \dot{\psi} + \frac{\sin \psi}{r} \right), \tag{A33}$$

$$p_r \equiv \frac{\partial L}{\partial \dot{q}_r} \equiv \frac{\partial L}{\partial \dot{r}} = \lambda_r, \tag{A34}$$

$$p_h \equiv \frac{\partial L}{\partial \dot{q}_h} \equiv \frac{\partial L}{\partial \dot{h}} = \lambda_h. \tag{A35}$$

As an alternative to the direct solution of **Equation A32** the extremal functions of **Equation A30** can also be obtained by solving the corresponding Hamilton equations. The Euler-Lagrange equations generally contain derivatives up to second order with the corresponding Hamilton equations only containing first-order derivatives (**Kibble and Berkshire, 2004**), which can make their (numerical) solution more straightforward. Furthermore, the Hamiltonian formalism is well suited for finding conserved quantities – that is, quantities that are constant with $s$ – and making use of them when analyzing the system at hand (**Kibble and Berkshire, 2004**). We follow here previous work on the arclength parametrization of membrane surfaces (**Deserno and Bickel, 2003**; **Deserno, 2004**; **Nowak and Chou, 2008**; **Zhang and Nguyen, 2008**; **Hashemi et al., 2014**; **Foret, 2014**) and determine the extremal functions of **Equation A30** by (numerically) solving the corresponding Hamilton equations. To this end, we note that the Hamitonian function associated with $q_\delta$ and $p_\delta$ is given by (**Kibble and Berkshire, 2004**)

$$H = p_\psi \dot{\psi} + p_r \dot{r} + p_h \dot{h} - L. \tag{A36}$$

Using **Equation A31** and **Equations A33–A35**, **Equation A36** yields (**Deserno, 2004**)

$$H = \frac{p_\psi^2}{4r} - p_\psi \frac{\sin \psi}{r} - \frac{2r}{\lambda^2}(1 - \cos \psi) + p_r \cos \psi + p_h \sin \psi. \tag{A37}$$

Note that, as in many (unforced) classical systems (**Kibble and Berkshire, 2004**), $H$ in **Equation A37** is a function of $q_\delta$ and $p_\delta$ only. In particular, $H$ in **Equation A37** does not have an explicit dependence on $s$, and $H$ is therefore conserved along $s$ (**Kibble and Berkshire, 2004**). The Hamilton equations are given by (**Kibble and Berkshire, 2004**)

$$\frac{\partial H}{\partial p_\delta} = \dot{q}_\delta, \quad \frac{\partial H}{\partial q_\delta} = -\dot{p}_\delta, \tag{A38}$$

which, for **Equation A37**, yields (**Deserno, 2004**)

$$\dot{\psi} = \frac{p_\psi}{2r} - \frac{\sin\psi}{r}, \tag{A39}$$

$$\dot{r} = \cos\psi, \tag{A40}$$

$$\dot{h} = \sin\psi, \tag{A41}$$

$$\dot{p}_\psi = \left(\frac{p_\psi}{r} - p_h\right)\cos\psi + \left(\frac{2r}{\lambda^2} + p_r\right)\sin\psi, \tag{A42}$$

$$\dot{p}_r = \frac{p_\psi}{r}\left(\frac{p_\psi}{4r} - \frac{\sin\psi}{r}\right) + \frac{2}{\lambda^2}(1 - \cos\psi), \tag{A43}$$

$$\dot{p}_h = 0. \tag{A44}$$

The solutions of **Equations A39–A44** specify the stationary shapes of Piezo's membrane footprint implied by **Equation A29**, from which the corresponding membrane deformation energy can be calculated (numerically) via **Equation A29** or via **Equation A30**.

## 2.2 Boundary conditions

**Equations A39–A44** must be solved subject to the boundary conditions set by the Piezo dome and the membrane compartment boundary. A first set of boundary conditions is obtained from simple geometric considerations. In particular, at the boundary of the Piezo dome and the surrounding membrane, we set

$$r(0) = R\sin\alpha, \tag{A45}$$

$$h(0) = -R\cos\alpha, \tag{A46}$$

which amounts to fixing the origin of the $r$-$h$ coordinate system (**Appendix 1—figure 1**). Having fixed the origin of the $r$-$h$ coordinate system via **Equations A45 and A46**, we take $r(s_0)$ and $h(s_0)$ to be free, that is, we assume that the values of $r(s_0)$ and $h(s_0)$ can be freely varied when finding the extremal functions of **Equation A30**. The corresponding 'natural' boundary conditions are given by (**Courant and Hilbert, 1953**)

$$\left.\frac{\partial L}{\partial \dot{r}}\right|_{s=s_0} = 0, \tag{A47}$$

$$\left.\frac{\partial L}{\partial \dot{h}}\right|_{s=s_0} = 0. \tag{A48}$$

From **Equation A48** with **Equation A31** it follows that $\lambda_h(s_0) = 0$, and **Equation A44** with **Equation A35** then yields $p_h = 0$ for $0 \leq s \leq s_0$, that is, the generalized momentum $p_h$ is conserved along $s$ and drops out of the problem. Using **Equations A31 and A34**, **Equation A47** can be rewritten as

$$p_r(s_0) = \lambda_r(s_0) = 0. \tag{A49}$$

A second set of boundary conditions encapsulates the key physical properties of the specific experimental setup under consideration. In particular, assuming that the tangents to the membrane profile change smoothly at the boundary of the Piezo dome and the surrounding membrane (**Appendix 1—figure 1**), we have

$$\psi(0) = \alpha, \tag{A50}$$

where $\alpha$ is given by **Equation A5**. Furthermore, we assume (**Deserno, 2004**), based on physical reasoning analogous to that behind **Equation A11**, that the tangent and curvature of the membrane profile in the $r$-$h$ plane are kept fixed at $s = s_0$,

$$\psi(s_0) = \beta, \tag{A51}$$

$$\dot{\psi}(s_0) = \mathcal{C}, \tag{A52}$$

where we use here the values $\beta = 0$ and $\mathcal{C} = 0$ corresponding to a flat membrane surface far enough away from Piezo.

Since $\beta = 0$ and $\mathcal{C} = 0$ in **Equations A51 and A52**, we expect from **Equation A33** that $p_\psi(s_0) = 0$ and, hence, **Equation A37** yields $H = p_r(s_0)$. (It is assumed here that $r(s_0)$ is finite.) Because $H$ in **Equation A37** is conserved along $s$, we have $H = p_r(s_0)$ for $0 \leq s \leq s_0$. The boundary condition in **Equation A49** thus implies $H = p_r(s_0) = 0$ for $0 \leq s \leq s_0$. At $s = 0$, we therefore have (**Deserno, 2004**)

$$p_r(0) = \frac{\tan \alpha}{R} \left\{ 1 + \frac{2R^2}{\lambda^2}(1 - \cos \alpha) - \left[R \dot{\psi}(0)\right]^2 \right\}, \tag{A53}$$

where we have used **Equation A37** with **Equations A33, A45, and A50**, and $\dot{\psi}(0)$ is unknown. Note that, for a given set of values of $r(0)$, $\psi(0)$, and $\dot{\psi}(0)$, the generalized momentum $p_\psi$ in **Equation A33** is completely determined at $s = 0$. Since **Equations A39–A44** are a set of first-order (coupled) ordinary differential equations, their solution is fixed by the 'initial' conditions $\psi(0)$, $r(0)$, $h(0)$, $p_\psi(0)$, $p_r(0)$, and $p_h(0)$, with the only unknown being $\dot{\psi}(0)$. A practical strategy for obtaining Piezo's membrane footprint through numerical solution of **Equations A39–A44** is therefore to generate a set of solutions corresponding to different values of $\dot{\psi}(0)$ and to select, among these solutions, the specific solution(s) satisfying **Equations A51 and A52** up to some numerical accuracy. Such a 'shooting' method (**Burden and Faires, 2011**; **Gautschi, 2012**) has been employed widely for numerically finding the stationary shapes implied by the arclength parametrization of **Equation A1** (**Peterson, 1985**; **Seifert et al., 1991**; **Jülicher and Seifert, 1994**; **Seifert, 1997**; **Deserno and Bickel, 2003**; **Deserno, 2004**; **Nowak and Chou, 2008**; **Zhang and Nguyen, 2008**; **Foret, 2014**; **Hashemi et al., 2014**; **Agudo-Canalejo and Lipowsky, 2016**; **Bahrami et al., 2016**), and we used the same approach here.

For completeness, we outline here the basic procedure for the solution of **Equations A39–A44** through the shooting method (**Deserno and Bickel, 2003**; **Deserno, 2004**; **Nowak and Chou, 2008**; **Zhang and Nguyen, 2008**; **Hashemi et al., 2014**; **Foret, 2014**). First, based on intuition gained from numerical experimentation, we chose a range of suitable $\dot{\psi}(0)$. More automated approaches for choosing $\dot{\psi}(0)$ are also available (**Burden and Faires, 2011**; **Gautschi, 2012**). Note that, to achieve a flat membrane profile far enough away from Piezo, we generally expect that $\dot{\psi}(0) < 0$. For the scenarios considered here we have that $-0.4 < \dot{\psi}(0) < 0$. Second, we solved **Equations A39–A44** subject to the initial conditions $\psi(0)$, $r(0)$, $h(0)$, $p_\psi(0)$, and $p_r(0)$ described above with $p_h = 0$ using standard numerical solvers of ordinary differential equations (**Wolfram Research, Inc., 2017**) up to some large (but necessarily finite) maximum value of the arclength. For each of these solutions, we numerically determined the smallest value of $s$ for which **Equations A51 and A52** are satisfied. (In practice, we only considered here **Equation A51** (**Deserno, 2004**).) For scenarios in which we assume that **Equations A51 and A52** hold only asymptotically for $s \to \infty$, we identified this value of $s$ with $s_0$ (**Deserno, 2004**), and selected among all the solutions corresponding to different $\dot{\psi}(0)$ the solution with the largest value of $s_0$. (It is assumed here that the exact solutions of **Equations A39–A44** do not oscillate about the boundary conditions in **Equations A51 and A52**.) For scenarios in which we assume that membrane compartmentalization favors solutions with a given value $s_0 = s_m$, we selected the solution with the smallest magnitude of $\psi(s_m)$, and confirmed for this solution that $s_m$ approximately corresponds to the smallest value of $s$ for which **Equation A51** is satisfied. (Again, it is assumed here that the exact solutions of **Equations A39–A44** do not oscillate about the boundary conditions in **Equation A51 and A52**.) The resulting solutions of **Equations A39–A44** specify the shape of the membrane footprint of the Piezo dome, from which we obtained the corresponding energy cost of Piezo-induced membrane shape deformations by numerically evaluating **Equation A30** over the integration domain $0 \leq s \leq s_0$ (**Wolfram Research, Inc., 2017**). We tested this numerical solution procedure (**Deserno and Bickel, 2003**; **Deserno, 2004**; **Nowak and Chou, 2008**; **Zhang and Nguyen, 2008**; **Hashemi et al., 2014**; **Foret, 2014**) for Piezo's membrane footprint by comparing the

membrane deformation energies implied by the Monge and arclength parameterizations of *Equation A1* (see Appendix 1-section 3).

## 3 Comparing arclength and Monge solutions

To check the numerical solution procedure described in Appendix 1-section 2 it is useful to compare numerical solutions obtained using the arclength parametrization of *Equation A1* with the corresponding exact analytical solutions obtained in Appendix 1-section 1 using the Monge parametrization of *Equation A1* (see *Appendix 1—figure 2*). For infinite (see *Appendix 1—figure 2A*) as well as finite (see *Appendix 1—figure 2B*) membrane compartment sizes, which we specify in terms of the arclength $s_m$ such that $0 \leq s \leq s_m$, we find that the numerical solutions obtained in the arclength parametrization of *Equation A1* agree with the corresponding analytical solutions obtained in the Monge parametrization of *Equation A1* for small enough membrane shape deformations (large enough $R$). In the case of infinite membrane compartments, the decrease in the relative difference of the analytical and numerical solutions with increasing $R$, shown in the lower panel of *Appendix 1—figure 2A*, is consistent with previous results (*Deserno and Bickel, 2003*). In the case of finite membrane compartments we find, in *Appendix 1—figure 2B*, a similar convergence of analytical and numerical results with increasing $R$, with a particularly small relative difference of the analytical and numerical solutions for $\gamma = 1.0\,k_B T/\mathrm{nm}^2$. This can be rationalized by noting that, for $\gamma = 1.0\,k_B T/\mathrm{nm}^2$ with $K_b = 20\,k_B T$, the characteristic decay length of Piezo-induced membrane shape deformations, $\lambda = \sqrt{K_b/\gamma} \approx 4.5$ nm, is of a comparable magnitude as the membrane compartment size $s_m = 5$ nm used in *Appendix 1—figure 2B*.

In *Appendix 1—figure 2B*, as well as the main text, we implemented finite membrane compartments by fixing the value of the maximum arclength $s_m$ such that $0 \leq s \leq s_m$ and imposing a flat membrane shape at the membrane compartment boundary. There are other ways of implementing finite membrane compartments in our physical model of the mechanical gating of Piezo. In particular, one may define the membrane compartment as having a fixed surface area, a fixed in-plane area, a fixed in-plane radius, or a fixed arclength. Furthermore, the boundary of the membrane compartment may locally impose different membrane shapes. To illustrate how a finite membrane compartment size can affect the mechanical gating of Piezo, we focused here on the particularly straightforward case of membrane compartments with a fixed arclength around the Piezo dome and a flat membrane shape at the membrane compartment boundary. Membrane compartments with other properties would yield different results. For instance, if the membrane shape at the membrane compartment boundary was chosen so as to locally match the shape of the corresponding solution obtained for an asymptotically flat, infinite membrane, a finite membrane compartment would yield, for $\gamma > 0$, an energy cost of Piezo's membrane footprint that is *decreased* compared to the associated energy cost of Piezo's membrane footprint for an asymptotically flat, infinite membrane. In contrast, we find in the main text (and also *Appendix 1—figure 2*) that, for a flat membrane shape at the membrane compartment boundary, the energy cost of Piezo's membrane footprint is *increased* in a finite membrane compartment compared to an asymptotically flat, infinite membrane.

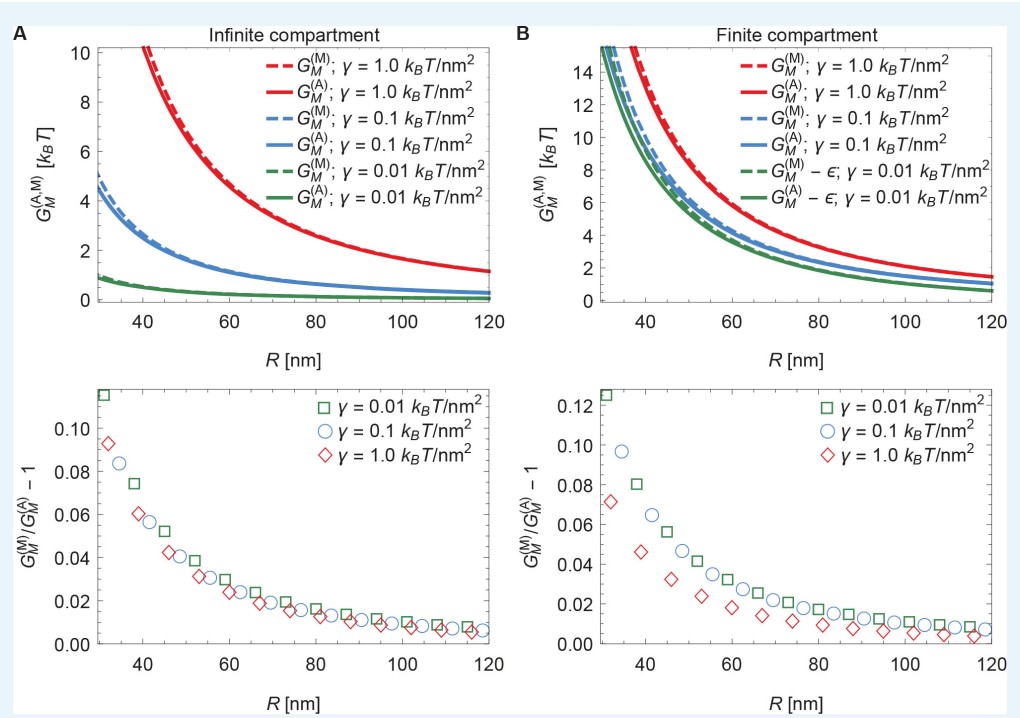

**Appendix 1—figure 2.** Comparison of arclength and Monge solutions. Energy cost of Piezo's membrane footprint $G_M$ as a function of the radius of curvature of the Piezo dome $R$ calculated numerically using the arclength parametrization of *Equation A1* (see Appendix 1-section 2) $[G_M^{(A)}]$ and analytically using the Monge parametrization of *Equation A1* (see Appendix 1-section 1) $[G_M^{(M)}]$ (upper panels), and corresponding relative difference of the analytical and numerical solutions (lower panels), for (**A**) infinite and (**B**) finite membrane compartments. We set $K_b = 20\,k_BT$ and $S_{cap} = 0.3 \times 4\pi \times 10.2^2\,\text{nm}^2 \approx 390\,\text{nm}^2$ (*Guo and MacKinnon, 2017*), and used the indicated values of the membrane tension. For (**B**) we used an unconstrained membrane arclength $s_m = 5\,\text{nm}$ separating the boundary of the Piezo dome and the boundary of the membrane compartment along the membrane in the radial direction. For ease of visualization, we shifted the curves corresponding to $\gamma = 0.01\,k_BT/\text{nm}^2$ by $\epsilon = 0.4\,k_BT$ in the upper panel of (**B**).

DOI: https://doi.org/10.7554/eLife.41968.011

The Monge parametrization of *Equation A1* fails to give quantitatively accurate results for the large membrane shape deformations implied by the observed Piezo dome structure with $R \approx 10.2\,\text{nm}$ (*Guo and MacKinnon, 2017*) (see *Appendix 1—figures 3–7*). In particular, the Monge parametrization of *Equation A1* overestimates the magnitude of the membrane shape deformations induced by the Piezo dome (*Appendix 1—figure 3*). Furthermore, the Monge parametrization of *Equation A1* yields contributions to the Piezo gating energy due to Piezo's membrane footprint, $\Delta G_M$, that are too large by a factor of approximately 4/3 to 12 depending on the specific scenario considered (*Appendix 1—figures 5 and 7*). However, we also find that the Monge parametrization of *Equation A1* gives the same qualitative results as the arclength parametrization of *Equation A1* for the scenarios considered in the main text (*Appendix 1—figures 3–7*).

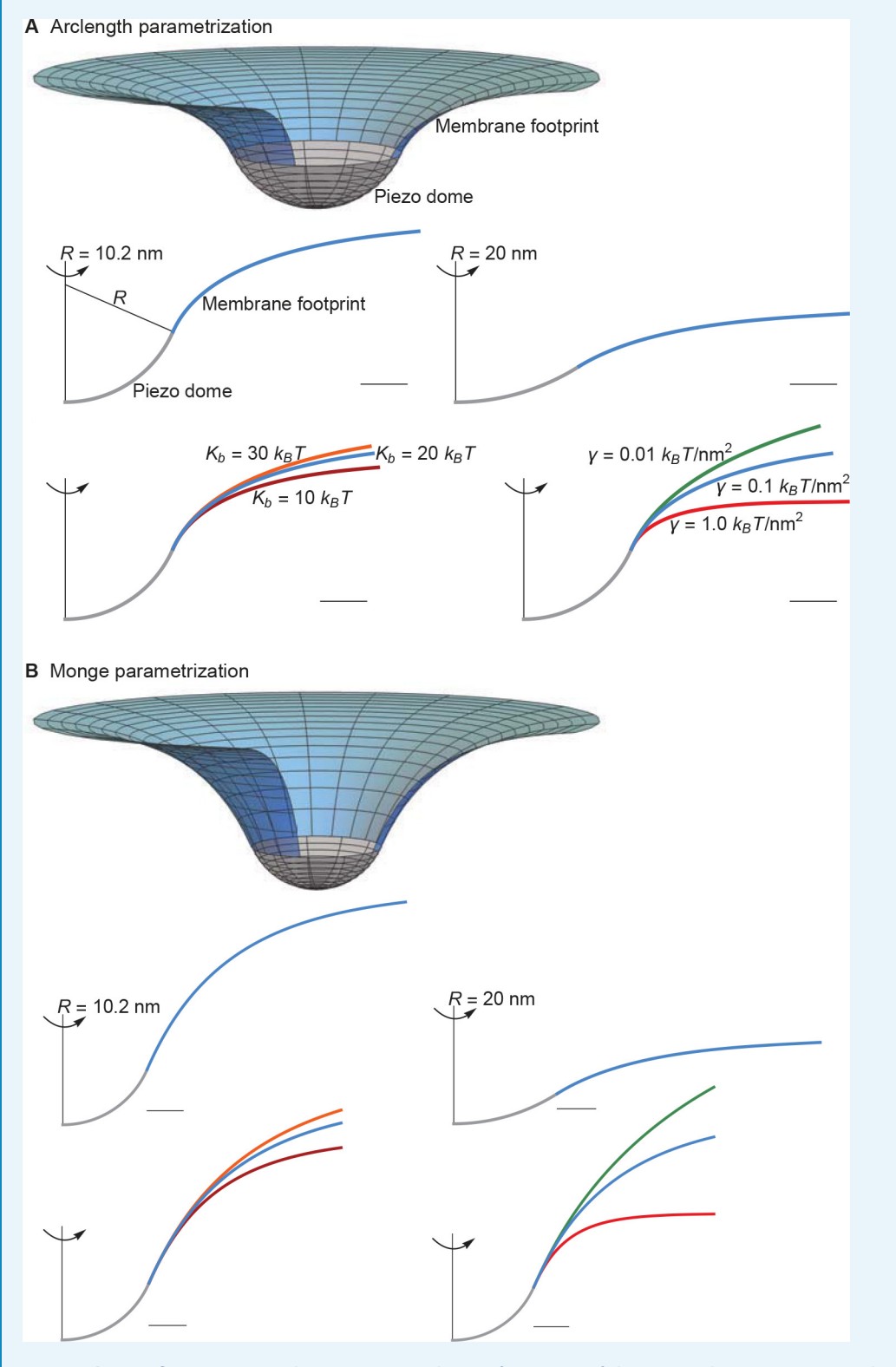

**Appendix 1—figure 3.** Supplement to membrane footprint of the Piezo dome. (**A**) Same plots as in *Figures 1C* and *2* of the main text, with the membrane footprints calculated numerically using the arclength parametrization of *Equation A1* (see Appendix 1-section 2) and (**B**) corresponding results with the membrane footprints calculated analytically using the Monge parametrization of *Equation A1* (see Appendix 1-section 1). We use the same labeling

conventions for (**A**) and (**B**). Scale bars, 4 nm. See *Figures 1C* and *2* of the main text for further details.

DOI: https://doi.org/10.7554/eLife.41968.012

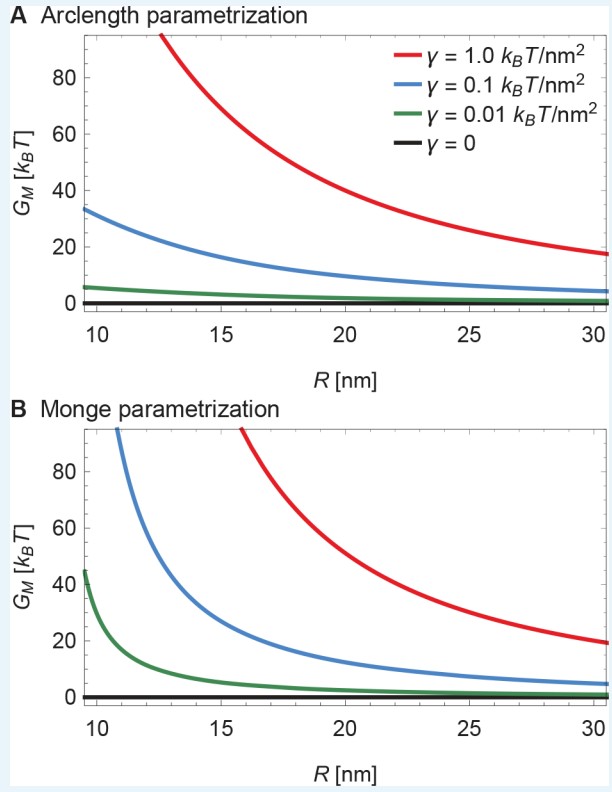

**Appendix 1—figure 4.** Supplement to energy of the Piezo membrane footprint. (**A**) Same plots as in *Figure 3A* of the main text, with $G_M$ calculated numerically using the arclength parametrization of *Equation A1* (see Appendix 1-section 2) and (**B**) corresponding results with $G_M$ calculated analytically using the Monge parametrization of *Equation A1* (see Appendix 1-section 1). We use the same labeling conventions for (**A**) and (**B**). See *Figure 3A* of the main text for further details.

DOI: https://doi.org/10.7554/eLife.41968.013

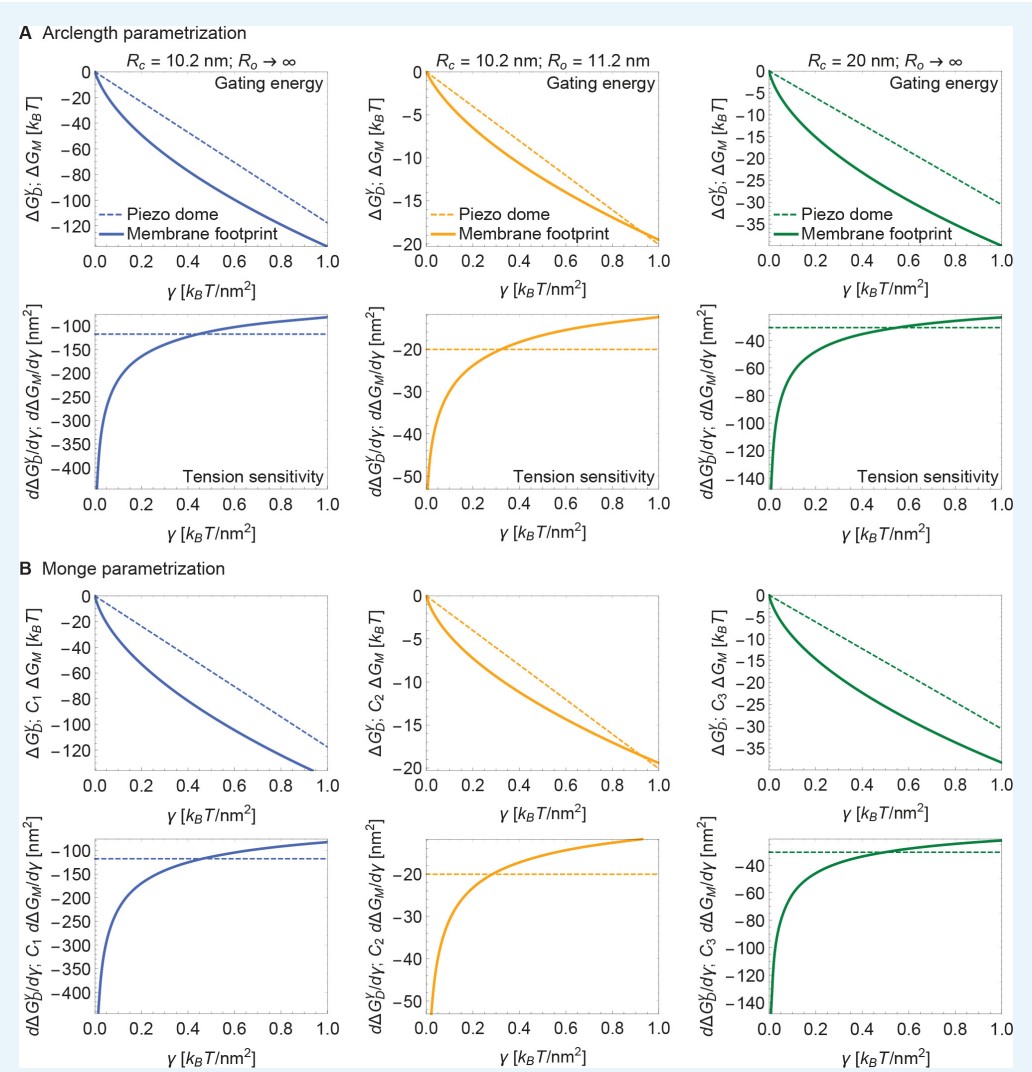

**Appendix 1—figure 5.** Supplement to energy of Piezo gating. (**A**) Same plots as in *Figure 4* of the main text, with $\Delta G_M$ calculated numerically using the arclength parametrization of *Equation A1* (see Appendix 1-section 2) and (**B**) corresponding results with $\Delta G_M$ calculated analytically using the Monge parametrization of *Equation A1* (see Appendix 1-section 1). We use the same labeling conventions for (**A**) and (**B**). For ease of visualization, we rescaled $\Delta G_M$ by $C_1 = 1/4$, $C_2 = 1/12$, or $C_3 = 3/4$ in (**B**) (left to right panels). See *Figure 4* of the main text for further details.

DOI: https://doi.org/10.7554/eLife.41968.014

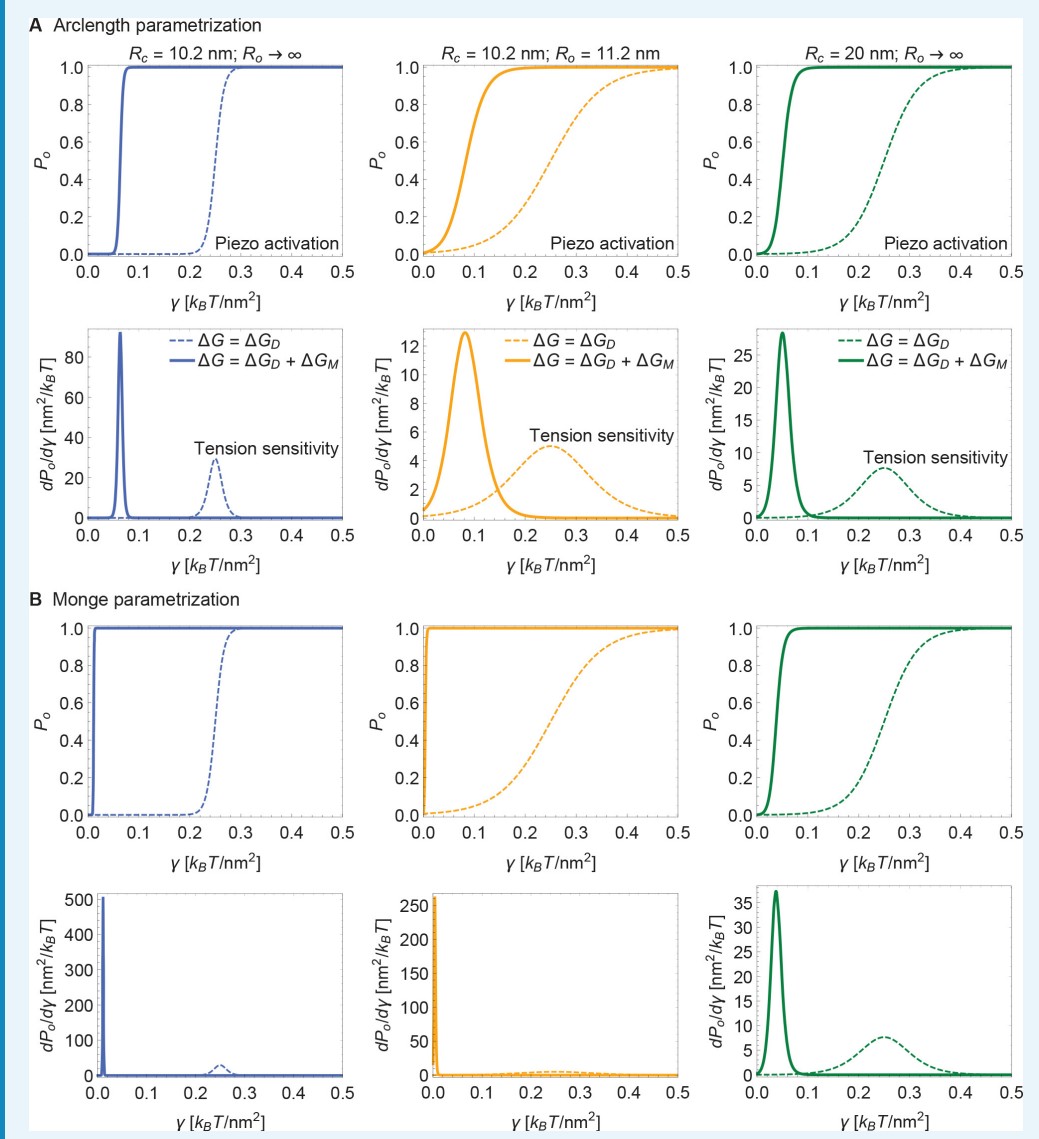

**Appendix 1—figure 6.** Supplement to Piezo activation through membrane tension. (**A**) Same plots as in *Figure 5* of the main text, with $\Delta G_M$ calculated numerically using the arclength parametrization of *Equation A1* (see Appendix 1-section 2) and (**B**) corresponding results with $\Delta G_M$ calculated analytically using the Monge parametrization of *Equation A1* (see Appendix 1-section 1). We employed the same values of $\Delta G_D^P$, and use the same labeling conventions, for (**A**) and (**B**). See *Figure 5* of the main text for further details.

DOI: https://doi.org/10.7554/eLife.41968.015

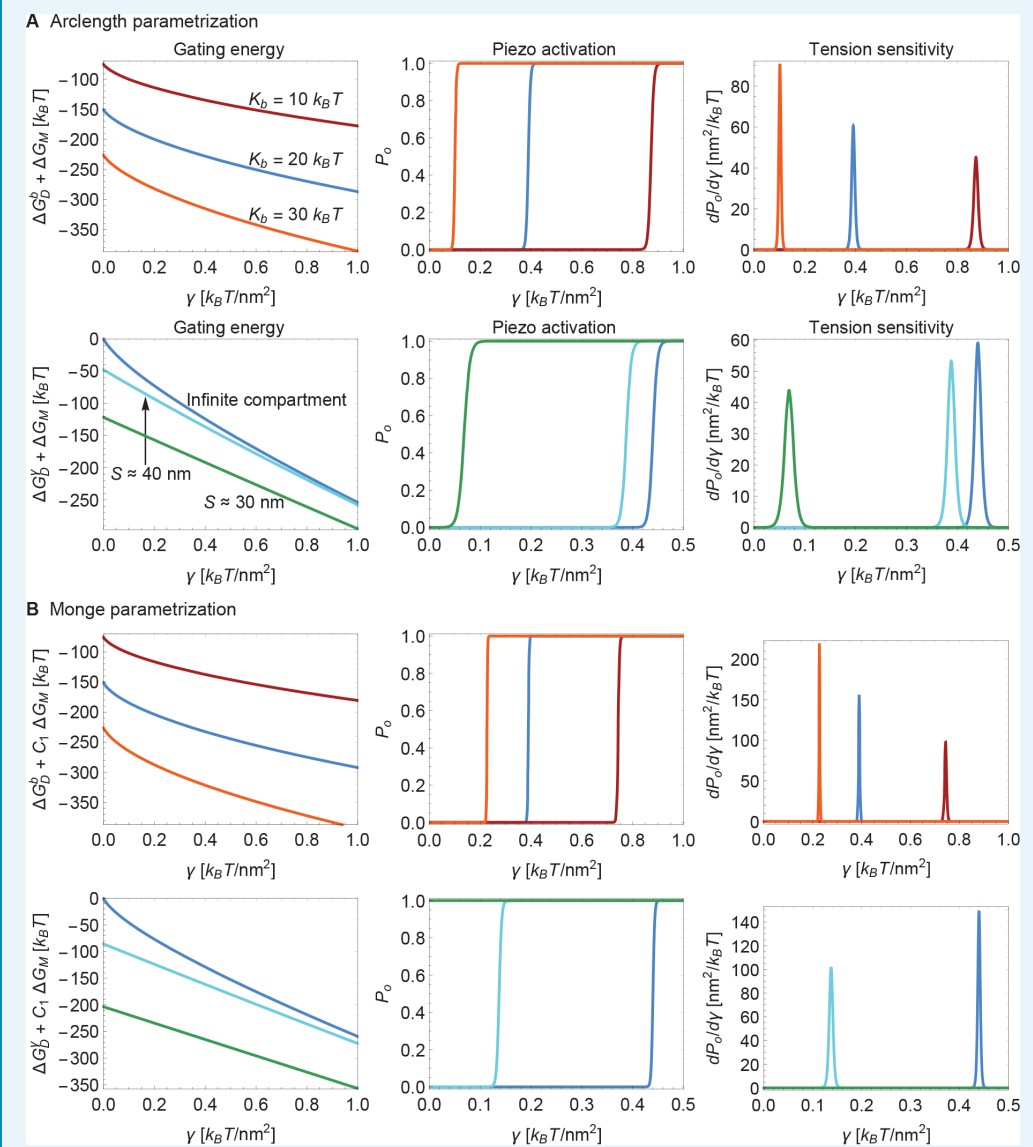

**Appendix 1—figure 7.** Supplement to modulation of Piezo gating through the membrane. (**A**) Same plots as in *Figure 6* of the main text, with $\Delta G_M$ calculated numerically using the arclength parametrization of *Equation A1* (see Appendix 1-section 2) and (**B**) corresponding results with $\Delta G_M$ calculated analytically using the Monge parametrization of *Equation A1* (see Appendix 1-section 1). We use the same labeling conventions for (**A**) and (**B**). For ease of visualization, we rescaled $\Delta G_M$ by $C_1 = 1/4$ in the left panels of (**B**). For (**B**) we set $\Delta G_D^P \approx 520\,k_BT$ (upper middle and upper right panels) and $\Delta G_D^P \approx 550\,k_BT$ (lower middle and lower right panels) for the (unknown) contribution of the protein energy to the Piezo gating energy such that gating occurs within the indicated tension ranges. See *Figure 6* of the main text for further details.

DOI: https://doi.org/10.7554/eLife.41968.016

