## [Decision Letter]

Thank you for submitting your article "Piezo's membrane footprint and its contribution to mechanosensitivity" for consideration by *eLife*. Your article has been reviewed by three peer reviewers, and the evaluation has been overseen by a Reviewing Editor and Richard Aldrich as the Senior Editor. The following individuals involved in review of your submission have agreed to reveal their identity: Ardem Patapoutian (Reviewer #1); Olaf S Andersen (Reviewer #2).

The reviewers have discussed the reviews with one another and the Reviewing Editor has drafted this decision to help you prepare a revised submission.

Summary:

This study explores the physical basis of mechanosensitivity of Piezo1 ion channels. Structural studies have shown that Piezo1 channels have a unique curved shape in detergent micelles. This curved architecture is largely retained in liposomes/lipid membranes such that the lipid membranes are significantly deformed upon incorporation of Piezo1 channels. This paper examines the importance of this membrane deformation on the overall gating and mechanosensitivity of the Piezo1 channels from an analytical standpoint. The overall conclusion is that Piezo, owing to its unusual shape, imposes a large structural perturbation (which the authors refer to as membrane footprint) on its surrounding membrane. This membrane footprint weighs such that the tension sensitivity of Piezo gating is greatest in the low-tension regime rendering Piezo poised to respond to the slightest changes in cell membrane tension. The authors' treatment of mechanosensitivity is elegant – it eschews details such as "structural domains" and "conformational changes" and yet is able to incorporate structural information to offer mechanistic insight into how Piezo channels work.

Essential revisions:

1) As the authors note, only partial molecular structures of Piezo have been determined. 12 predicted TM helices per promoter are conspicuously absent, which could contribute significantly to the shape of Piezo. Do the authors think the additional TMs extend the Piezo "dome" or are they part of the membrane footprint? How might the additional TMs affect the parameters in the mechanical calculations?

2) The authors only consider a single aspect observed in the Piezo structure – its unique curvature – in constructing and describing the model of mechanosensitivity. In the context of the dome/membrane footprint gating models, what other aspects of the structure could be important? For example, as seen in Figure 1B, the "arms" of Piezo are "tucked" inward, creating a relatively compact triskelion. Might the three armed architecture afford the ability of Piezo to "curl" outward in an expanded/open state?

3) What would be the curvature of Piezo1, in the open state, which would enable it to be sensitive to mechanical stimuli? This might allow one to rationalize the pretty large (~47kT) energy difference between open and closed states (used for many of the simulations) presented.

4) Would the authors predict that Piezo1 is osmosensitive as well? Change in osmotic pressure would change the intrinsic curvature of the membrane and thus the membrane deformation energy which according to the authors' model would affect gating and mechanosensitivity of the channel. To the best of my knowledge, Piezo1 channels show very little osmosensitivity. It might serve the paper if the authors briefly discuss the different modes of mechanosensitivity (osmosensitive, shear-sensitive, mechano- or stretch sensitive, etc.) and how their model relates to Piezo1's unique mode of mechanical response.

5) Two related points raised by the reviewers:

Throughout the paper the authors consider the Piezo-membrane system as consisting of a single Piezo trimer and its surrounding membrane, and that the "membrane footprint" of Piezo can extend tens of nanometers beyond the actual surface area occupied by the protein. But cellular membranes are highly heterogenous and is occupied by many membrane proteins, some of which may interact with Piezo and modulate function. It seems that the deformed membrane surrounding Piezo would be energetically unfavorable to be occupied by other membrane proteins, at least those that prefer planar membranes. A further consequence of the membrane footprint model is that two neighboring Piezo molecules would repel each other to prevent overlap of deformation footprints (Philips, Ursell et al., Nature 2009). Can the authors discuss how the membrane footprint might effect interaction of Piezos with other membrane proteins, and whether their model precludes the formation of Piezo clustering (a behavior seen in various other ion channel families)?

The final question which I have is whether the authors think that if their proposed mechanism is 'correct' and the membrane deformation energy is a crucial component of gating energetics, would there be interchannel cooperativity – i.e. cooperativity between (two or more) neighboring channels? The authors describe the membrane footprint of a single channel and how it relates to its gating. However, if there are two or more channels in the vicinity, then the overall membrane footprint of the multiple channels might (and in all likelihood will) be different from the sum of individual channels and therefore the energetics and mechanosensitivity would also be different. Furthermore, the nature of the membrane (intrinsic curvature and composition) would also in likelihood affect such cooperativity. I would think that this is an important point and wonder what the authors think about it. It might be worthwhile to discuss this aspect, even if briefly.

---

## [Author Response]

Essential revisions:1) As the authors note, only partial molecular structures of Piezo have been determined. 12 predicted TM helices per promoter are conspicuously absent, which could contribute significantly to the shape of Piezo. Do the authors think the additional TMs extend the Piezo "dome" or are they part of the membrane footprint? How might the additional TMs affect the parameters in the mechanical calculations?

The presence of the additional TMs would make the arms longer and the area of Piezo larger. However, because the first 12 helices corresponding to the first 3 domains are not visualized in any of the structures that have been solved, we suspect that there is likely a flexible connection between the first 3 domains and the others, so that the first 3 can adopt many orientations with respect to the remainder. If this is the case, then they would be structurally passive and not contribute to the formation of the dome (i.e., they might extend into the membrane footprint but presumably would not affect its shape much).

In summary, the dome – defined geometrically by its area and radius of curvature – is formed by the rigid components of Piezo. The first 3 domains do not seem rigidly attached to Piezo. If, in a particular conformation of the channel, the first 3 domains adopt a more rigidly attached configuration, then they could increase the area and alter the energetics of Piezo. For now, the model is based on what we have been able to see.

2) The authors only consider a single aspect observed in the Piezo structure – its unique curvature – in constructing and describing the model of mechanosensitivity. In the context of the dome/membrane footprint gating models, what other aspects of the structure could be important? For example, as seen in Figure 1B, the "arms" of Piezo are "tucked" inward, creating a relatively compact triskelion. Might the three armed architecture afford the ability of Piezo to "curl" outward in an expanded/open state?

Viewed down the 3-fold axis the arms are indeed tucked inward. Viewed from other angles one can see that each arm adopts a left-handed helical twist such that it curls around an approximately spherical dome-shaped membrane surface. In other words, curving of the arms is actually consistent with a spherical dome-like shape. If one imagines that the arms remain adherent to the membrane surface as the dome decreases its radius of curvature (i.e., flattens), then the arms would indeed straighten out (i.e., become less tucked inward) as the dome flattens. Thus, straightening of the arms, at least to some extent, is expected.

We want to emphasize, however, that the spherical dome model is an approximation of some more complex geometry. This approximation captures the essence of a system that we think works by pulling membrane area out of the membrane plane in the closed conformation and permitting that area to re-enter the membrane plane upon opening (i.e., by flattening). In detail, aspects of the structure of Piezo that cause deviations from the spherical dome shape (e.g., by breaking rotational symmetry about the central pore axis of Piezo) will perturb the shape of the membrane footprint and the magnitude of the energy. Nevertheless, the membrane footprint will contribute to the energetics of channel gating.

To highlight this important point raised, we have added emphasis in the third paragraph of the subsection “Shape and energy of the membrane footprint” that the spherical dome is an approximation. We further address this point in the last paragraph of the aforementioned subsection and in the third paragraph of the Discussion section.

3) What would be the curvature of Piezo1, in the open state, which would enable it to be sensitive to mechanical stimuli? This might allow one to rationalize the pretty large (~47kT) energy difference between open and closed states (used for many of the simulations) presented.

This question was addressed in the calculation of membrane shapes and energies for three different gating scenarios in Figures 4 and 5. For the closed and opened radius of curvature, Rc and Ro respectively, we plot energy (Figure 4) and open probability (Figure 5) – and their slopes (sensitivity) – for values Rc→Ro (nm) corresponding to 10.2→∞, 10.2→11.2, and 20→∞. Thus, a change in radius of curvature as small as 10.2→11.2 nm (Figures 4B and 5B), for example, will in theory render Piezo quite sensitive to membrane tension. In Figure 5, the unknown protein energy difference between closed and open states (ΔGDP) was selected for each gating geometry so as to place the tension curve within the ranges shown (similar to experimental observation). In other words, we do show that the curvature changes rationalize those large energies. Incidentally, all of the analysis presented here entailed analytic or numerical calculations – in particular, solution of the Euler-Lagrange or Hamilton equations associated with equation 1 – rather than simulations.

We have revised our manuscript to emphasize, in the second paragraph of the subsection “Influence of the membrane footprint on gating”, the rationale behind the gating geometries considered in Figures 4 and 5.

4) Would the authors predict that Piezo1 is osmosensitive as well? Change in osmotic pressure would change the intrinsic curvature of the membrane and thus the membrane deformation energy which according to the authors' model would affect gating and mechanosensitivity of the channel. To the best of my knowledge, Piezo1 channels show very little osmosensitivity. It might serve the paper if the authors briefly discuss the different modes of mechanosensitivity (osmosensitive, shear-sensitive, mechano- or stretch sensitive, etc.) and how their model relates to Piezo1's unique mode of mechanical response.

Whether osmotic pressure is predicted by the model to activate Piezo1 depends on the scenario considered. In short, to the extent that an osmotic pressure difference across a cell membrane produces membrane unfolding and cell swelling – a regime in which the Young-Laplace equation will apply – the model predicts that an osmotic pressure difference will activate the channel through increases in membrane tension. That said, eukaryotic cells seem to have many compensatory mechanisms that minimize increases in membrane tension over a slow timescale. For example, caveolea appear to serve as membrane area buffers, allowing the membrane to ‘unfold’ when required. Thus, whether or not Piezo1 opens would seem to depend on the timescale over which a tension-inducing membrane perturbation occurs relative to the timescale of compensatory mechanisms (i.e., membrane unfolding, ion and water transport, etc.). To speculate with a specific example, rapid poking of the cell membrane opens Piezo1 channels (transiently), but an osmotic stress that would take more time to evolve might not. In summary, experimental tests to see whether Piezo1 can be activated by an osmotic pressure difference have to ensure that the Young-Laplace regime is achieved.

As alluded to by the reviewer, chemical differences between the two lipid bilayer leaflets of the membrane may induce a spontaneous curvature of the membrane. We have revised our manuscript to discuss, in the first paragraph of the subsection “Shape and energy of the membrane footprint”, how spontaneous curvature of the membrane would enter our model of Piezo’s membrane footprint.

5) Two related points raised by the reviewers:Throughout the paper the authors consider the Piezo-membrane system as consisting of a single Piezo trimer and its surrounding membrane, and that the "membrane footprint" of Piezo can extend tens of nanometers beyond the actual surface area occupied by the protein. But cellular membranes are highly heterogenous and is occupied by many membrane proteins, some of which may interact with Piezo and modulate function. It seems that the deformed membrane surrounding Piezo would be energetically unfavorable to be occupied by other membrane proteins, at least those that prefer planar membranes. A further consequence of the membrane footprint model is that two neighboring Piezo molecules would repel each other to prevent overlap of deformation footprints (Philips, Ursell et al., Nature 2009). Can the authors discuss how the membrane footprint might effect interaction of Piezos with other membrane proteins, and whether their model precludes the formation of Piezo clustering (a behavior seen in various other ion channel families)?The final question which I have is whether the authors think that if their proposed mechanism is 'correct' and the membrane deformation energy is a crucial component of gating energetics, would there be interchannel cooperativity – i.e. cooperativity between (two or more) neighboring channels? The authors describe the membrane footprint of a single channel and how it relates to its gating. However, if there are two or more channels in the vicinity, then the overall membrane footprint of the multiple channels might (and in all likelihood will) be different from the sum of individual channels and therefore the energetics and mechanosensitivity would also be different. Furthermore, the nature of the membrane (intrinsic curvature and composition) would also in likelihood affect such cooperativity. I would think that this is an important point and wonder what the authors think about it. It might be worthwhile to discuss this aspect, even if briefly.

To the first point, yes, Piezo’s membrane footprint should influence the distribution of molecules – both lipids and proteins – in the surrounding membrane. In particular, Piezo’s membrane footprint should attract lipids and proteins that exhibit an energetic preference for the curved shape of the membrane footprint, and repel molecules that “prefer” other membrane shapes. Thus, a prediction of the model is that molecules that couple to the membrane shape should show a distribution in the vicinity of Piezo that is different from their bulk distribution. Experiments to test this prediction are underway.

The specific case of Piezo surrounding Piezo is queried. Yes, Piezo channels should interact with nearby Piezo channels through their respective membrane footprints. We agree with the reviewers and with the cited reference that Piezo pairs, it would seem, should repel each other. The possible influence that groupings of Piezo could have on each other is more complex and an interesting topic for further study. We note also that clustering of membrane proteins is often mediated by cytoplasmic anchoring proteins.

The second point is a corollary of the first. Just as Piezo should influence the composition of its surrounding membrane, the composition of the surrounding membrane should influence the energetics of Piezo gating. That is to say, lipids or other proteins that integrate favorably into Piezo’s curved membrane footprint should alter the energetics of the footprint and thus alter the functional behavior of Piezo. This raises interesting possibilities for the regulation of Piezo gating in different membrane environments. The reviewers raise the special case of the ‘other protein’ being Piezo. Yes, we would predict that Piezo channels at a certain density should influence each other’s gating. Experiments to test this prediction are also underway.

We have added these important aspects to the second paragraph of the subsection “Modulation of gating through the membrane”.